# DIVERSE DICTIONARY LEARNING

**Yujia Zheng**[1]    **Zijian Li**[2]    **Shunxing Fan**[2]    **Andrew Gordon Wilson**[3]    **Kun Zhang**[1,2]
[1] CMU    [2] MBZUAI    [3] NYU

## ABSTRACT

Given only observational data $X = g(Z)$, where both the latent variables $Z$ and the generating process $g$ are unknown, recovering $Z$ is ill-posed without additional assumptions. Existing methods often assume linearity or rely on auxiliary supervision and functional constraints. However, such assumptions are rarely verifiable in practice, and most theoretical guarantees break down under even mild violations, leaving uncertainty about how to reliably understand the hidden world. To make identifiability *actionable* in the real-world scenarios, we take a complementary view: in the general settings where full identifiability is unattainable, *what can still be recovered with guarantees*, and *what biases could be universally adopted*? We introduce the problem of *diverse dictionary learning* to formalize this view. Specifically, we show that intersections, complements, and symmetric differences of latent variables linked to arbitrary observations, along with the latent-to-observed dependency structure, are still identifiable up to appropriate indeterminacies even without strong assumptions. These set-theoretic results can be composed using set algebra to construct structured and essential views of the hidden world, such as *genus-differentia* definitions. When sufficient structural diversity is present, they further imply full identifiability of all latent variables. Notably, all identifiability benefits follow from a simple inductive bias during estimation that can be readily integrated into most models. We validate the theory and demonstrate the benefits of the bias on both synthetic and real-world data.

## 1 INTRODUCTION

Dictionary learning, in its most general form, assumes that observations $X$ are generated by latent variables $Z$ through an unknown function $f$, i.e., $X = f(Z)$. The goal is to recover the latent generative process from observational data, a fundamental task in both science and machine learning. The nonparametric formulation $X = f(Z)$ unifies a wide range of latent variable models, including independent component analysis, factor analysis, and causal representation learning.

Identifiability, the ability to recover the true generative model from data, is crucial for understanding the hidden world. Yet in general dictionary learning, the problem is fundamentally ill-posed without additional assumptions, akin to finding a needle in a haystack. To reduce this ambiguity, most prior work imposes strong parametric constraints to limit the potential solution space. This practice is so widespread that, although dictionary learning is fundamentally nonparametric, it is almost always instantiated as a linear model, where observations are sparse linear combinations of latent variables (Olshausen & Field, 1997; Aharon et al., 2006; Geadah et al., 2024). However, this linearity could be overly restrictive and fails to capture the complexity of many real-world generative processes. A relevant example is sparse autoencoders (SAEs), commonly used in mechanistic interpretability, especially for foundation models. Although effective in some settings, SAEs are rooted in sparse linear dictionary learning, raising concerns about their ability to represent the inherently nonlinear structure of large-scale neural representations.

Many efforts have been made to relax the linearity assumption. In nonlinear ICA, one line of work leverages auxiliary variables as weak supervision to achieve identifiability under statistical independence (Hyvärinen & Morioka, 2016; Hyvärinen et al., 2019; Yao et al., 2021; Hälvä et al., 2021; Lachapelle et al., 2022), while another constrains the mixing function itself (Taleb & Jutten, 1999; Moran et al., 2021; Kivva et al., 2022; Zheng et al., 2022; Buchholz et al., 2022). In causal representation learning, identifiability often depends on access to interventional data (von Kügelgen et al., 2023; Jiang & Aragam, 2023; Jin & Syrgkanis, 2023; Zhang et al., 2024) or counterfactual

views (Von Kügelgen et al., 2021; Brehmer et al., 2022), which assume some control over the data-generating process to enable meaningful manipulation.

However, a gap remains between theoretical guarantees and practical utility. Theoretically, while additional assumptions can yield recovery guarantees, it is rarely possible to verify whether such assumptions hold in practice. Understanding what guarantees remain valid under assumption violations is therefore essential for reliably uncovering the truth in general settings. Practically, we are less concerned with identifiability under ideal conditions and more interested in which inductive biases promote recovery, especially when the ground truth is unknown. Yet most existing approaches fail to offer any guarantees under even mild violations of their assumptions, making their associated biases, such as contrastive objectives or weak supervision, difficult to generalize across settings.

Therefore, in **the general scenarios**, two questions remain:

- *What aspects of the latent process can still be recovered?*
- *What inductive biases should be introduced to guide recovery?*

To answer these questions and thus achieve *actionable* identifiability, we focus on a new problem aiming to offer meaningful guarantees across a wide range of scenarios: *diverse dictionary learning*. Rather than seeking to recover all latent variables in the system, we consider a complementary question: what aspects of the latent process remain identifiable even in the general settings with only basic assumptions? We show that, even without specific parametric constraints or auxiliary supervision, structured subsets of latent variables can still be identified through their set-theoretic relationships with observed variables. In particular, for any set of observed variables, the intersection, complement, and symmetric difference of their associated latent supports are identifiable (Thm. 1). Moreover, the dependency structure between latent and observed variables is also identifiable up to standard indeterminacy of relabeling (Thm. 2). These flexible results naturally uncover many informative perspectives of the hidden world through the lens of *diversity*: the intersection captures the common latent factors (*genus*) underlying multiple objects, while the complement and symmetric difference allow us to isolate the parts that are unique or non-overlapping (*differentia*), providing a principled way to understand the hidden world from the classical *genus-differentia* definitions (Granger, 1984) (Prop. 1) or the atomic regions in the Venn diagram (Sec. 3.2).

Since this form of identifiability is defined entirely through basic set-theoretic operations, it is highly flexible and applies to arbitrary subsets of observed variables based on set algebra. When the full set of observed variables is considered and the dependency structure between latent and observed variables is sufficiently *diverse*, it becomes possible to recover all latent variables, yielding a generalized structural criterion for full identifiability (Thm. 3). Notably, for estimation, these identifiability benefits require only a simple sparsity regularization on the dependency structure, which can be readily implemented in most models that admit a Jacobian. Our theory also makes it rather universal, supporting meaningful recovery across a wide range of settings, from partial to full identifiability, and thus serves as a robust and broadly applicable regularization principle. We incorporate this universal bias into different types of generative models and observe immediate benefits from the corresponding identifiability guarantee in both synthetic and real-world datasets.

## 2 BACKGROUND AND PROBLEM SETUP

We adopt the standard perspective of latent variable models, where the observed world is generated from latent variables through a hidden process:

$$X = g(Z), \tag{1}$$

where $X = (X_1, \cdots, X_{d_x}) \in \mathbb{R}^{d_x}$ denotes the observed variables, and $Z = (Z_1, \cdots, Z_{d_z}) \in \mathbb{R}^{d_z}$ denotes the latent variables. Let $\mathcal{X}$ and $\mathcal{Z}$ denote the supports of $X$ and $Z$, respectively.

**Connection to linear dictionary learning.** Our task can be viewed as a nonlinear version of classical dictionary learning. Both classical approaches (Olshausen & Field, 1997; Aharon et al., 2006) and more recent ones (Hu & Huang, 2023; Sun & Huang, 2025) model observations as **linear** combinations of dictionary atoms $D$, i.e., $X = DZ$. Differently, we consider the **nonlinear** setting $X = g(Z)$, where $g$ is a nonlinear function. Although arising in different contexts, linear dictionary learning provides a useful analogy for some necessary conditions to avoid ill-posed settings. In the linear case, conditions like Restricted Isometry Property had to be introduced, which ensure that

different latent codes map to distinguishable outputs, making the linear operator injective and thus no information is lost (Foucart & Rauhut, 2013; Jung et al., 2016). By analogy, the nonlinear setting also requires restrictions on $g$ to ensure injectivity. Following the literature on nonlinear identifiability, $g$ is assumed to be a $\mathcal{C}^2$ diffeomorphism onto its image (smooth and injective) (Hyvärinen & Pajunen, 1999; Lachapelle et al., 2022; Hyvärinen et al., 2024; Moran & Aragam, 2025).

**Connection to nonlinear identifiability results.** However, simply avoiding information loss is insufficient for full latent recovery with guarantees in the nonlinear regime. Prior work (see the survey (Hyvärinen et al., 2024)) addresses this by constraining the form of $g$ (e.g., post-nonlinear models) or by introducing auxiliary information, such as domain or time indices, or interventional/counterfactual data. In contrast, we focus on general real-world settings and deliberately avoid such assumptions, aiming to understand what can be recovered from this minimal setup. Naturally, some basic conditions, such as invertibility and differentiability, are necessary to rule out pathological cases, but our goal is to keep these as general as possible, even with the trade-off that recovering every latent variable becomes infeasible.

**Remark 1** (Extension to noisy processes). *Equation* (1) *considers a deterministic function, but it can be naturally extended to settings with additive noise using standard deconvolution (Kivva et al., 2022), or to more general noise models under additional assumptions (Hu & Schennach, 2008).*

**Structure.** The dependency structure between latent and observed variables, though hidden, captures the fundamental relationships underlying the data and is inherently nonparametric. To explore theoretical guarantees in general settings, this structure provides a natural starting point (Moran et al., 2021; Zheng et al., 2022; Kivva et al., 2022). Before diving deep into the hidden relations, we need to formalize them from the nonparametric functions. We first define the nonzero pattern of a matrix-valued function as:

**Definition 1.** *The **support** of a matrix-valued function $\mathbf{M} : \Theta \to \mathbb{R}^{m \times n}$ is the set of index pairs $(i, j)$ such that the $(i, j)$-th entry of $\mathbf{M}(\theta)$ is nonzero for some input $\theta \in \Theta$:*

$$\mathrm{supp}(\mathbf{M}; \Theta) \coloneqq \{(i, j) \in [m] \times [n] \,|\, \exists \theta \in \Theta, \mathbf{M}(\theta)_{i,j} \neq 0\}.$$

Figure 1: Example of structure.

For a constant matrix, its support is a special case of Defn. 1, which is the set of indices of non-zero elements. Then, we define the dependency structure as the support of the Jacobian of $g$:

**Definition 2.** *The **dependency structure** between latent variables $Z$ and observed variables $X = g(Z)$ is defined as the support of the Jacobian matrix of $g$. Formally,*

$$\mathcal{S} \coloneqq \mathrm{supp}(D_z g; \mathcal{Z}) = \left\{(i, j) \in [d_x] \times [d_z] \,\middle|\, \exists z \in \mathcal{Z}, \frac{\partial g_i(z)}{\partial z_j} \neq 0\right\}.$$

This structure $\mathcal{S}$ captures which latent variables functionally influence which observed variables through the generative map $g$. It might be noteworthy that, since it is defined via the Jacobian, it reflects functional rather than statistical dependencies. In particular, it does not require statistical independence of $Z$ and is therefore not limited to the mixing structures typically considered in ICA.

**Example 1.** *Figure 1 illustrates the dependency structure of a generative process. The top panel shows the ground-truth mapping from latent variables $Z = (Z_1, Z_2, Z_3)$ to observed variables $X = (X_1, X_2, X_3)$. The bottom panel shows the support of the Jacobian $D_Z g(Z)$, where non-zero entries are marked with "$*$". Notably, the Jacobian structure also captures dependencies between latent variables, such as the interaction between $Z_1$ and $Z_2$.*

## 3 THEORY

In this section, we develop the identifiability theory of diverse dictionary learning. Our theory begins with a generalized notion of identifiability based on set-theoretic indeterminacy (Sec. 3.1), capturing what remains recoverable under minimal assumptions. We then illustrate its practical implications, such as disentanglement and atomic region recovery, through concrete examples (Sec. 3.2). These insights motivate the formal guarantees in Thms. 1 and 2 (Sec. 3.3). Finally, we show how the same framework extends naturally to element-wise identifiability under a generalized structural condition (Thm. 3, Sec. 3.4). *All proofs are provided in Appx. A.*

### 3.1 CHARACTERIZATION OF GENERALIZED IDENTIFIABILITY

As previously discussed, identifying all latent variables is fundamentally ill-posed without additional information, such as restricted functional classes or multiple distributions. In general scenarios where such constraints are absent, a natural question arises: what aspects of the latent process remain recoverable? Before presenting our identifiability results, we first formalize this goal, which has not been addressed in the existing literature.

We begin by defining when two models are observationally indistinguishable from the perspective of the observed data, which is the goal of estimation based on observation.

**Definition 3** (**Observational equivalence**). *we say there is an **observational equivalence** between two models $\theta = (g, p_Z)$ and $\hat{\theta} = (\hat{g}, p_{\hat{Z}})$, denoted $\theta \sim_{obs} \hat{\theta}$, if and only if,*

$$p(x; \theta) = p(x; \hat{\theta}), \quad \forall x \in \mathcal{X}.$$

Given that estimation yields an observationally equivalent model, our goal is to determine whether the latent variables recovered by this model correspond meaningfully to those in the ground-truth model. Since we avoid placing restrictive assumptions on the entire system, we adopt a localized perspective: instead of analyzing global correspondence, we examine the relationship between latent components at the level of specific observed variables. Inspired by set theory, we introduce a new notion of indeterminacy that formalizes ambiguity through basic set-theoretic operations.

**Definition 4** (**Latent index set**). *For any set of observed variables $X_S$, its latent index set $I_S \subseteq [d_z]$ is defined as*

$$I_S := \{\, i \in [d_z] \mid \tfrac{\partial X_S}{\partial Z_i} \neq 0 \,\},$$

*i.e., the set of indices of latent variables $Z_{I_S}$ that influence $X_S$.*

**Definition 5** (**Set-theoretic indeterminacy**). *There is a **set-theoretic indeterminacy** between two models $\theta = (g, p_Z)$ and $\hat{\theta} = (\hat{g}, p_{\hat{Z}})$, denoted $\theta \sim_{set} \hat{\theta}$, if and only if, for any two sets of observed variables $X_K$ and $X_V$, and their latent index sets $I_K$ and $I_V$, there exists a permutation $\pi$ over $[d_z]$ such that $Z_i$ is not a function of $\hat{Z}_{\pi(j)}$[1] for all $(i, j)$ satisfying at least one of the following:*

*(i) (Intersection) $i \in I_K \cap I_V$, $j \in I_K \Delta I_V$;*

*(ii) (Symmetric difference[2]) $i \in I_K \Delta I_V$, $j \in I_K \cap I_V$;*

*(iii) (Complement) $i \in I_K \setminus I_V$, $j \in I_V \setminus I_K$, or $i \in I_V \setminus I_K$, $j \in I_K \setminus I_V$.*

Intuitively, set-theoretic indeterminacy guarantees that certain components of the latent variables defined by basic set-theoretic operations are disentangled from the rest, with an example as:

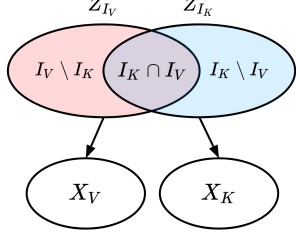

Figure 2: Example of Defn. 5.

**Example 2.** *Figure 2 illustrates latent variables indexed by $I_K$ and $I_V$, which influence the observed variable sets $X_K$ and $X_V$. The intersection $I_K \cap I_V$ contains shared latent factors, while the symmetric difference $I_K \Delta I_V$ consists of those unique to one set but not the other. According to set-theoretic indeterminacy, latent variables in the intersection $I_K \cap I_V$ cannot be expressed as functions of those in the symmetric difference $I_K \Delta I_V$, ensuring that shared components remain disentangled from exclusive ones. Similarly, variables in the symmetric difference $I_K \Delta I_V$ cannot be entangled with $I_K \cap I_V$, preserving directional separability. Finally, the complement condition prohibits mutual entanglement between the exclusive parts $I_K \setminus I_V$ and $I_V \setminus I_K$, guaranteeing that what is unique to one observed group cannot explain what is unique to another.*

Because these operations form the foundation of set algebra, they can be flexibly composed to derive a variety of meaningful perspectives on the hidden variables, which we will detail later. We are now ready to define what it means for a model to have generalized identifiability.

---

[1]Given the standard invertibility assumption, the reverse also holds because of the block-diagonal Jacobian, which applies similarly in related definitions.

[2]The symmetric difference $I_K \Delta I_V$ denotes elements in $I_K$ or $I_V$ but not in both, i.e., $(I_K \setminus I_V) \cup (I_V \setminus I_K)$.

**Definition 6** (**Generalized identifiability**). *For a model $\theta = (g, p_Z)$, we have **generalized identifiability**, if and only if, for any other model $\hat{\theta} = (\hat{g}, p_{\hat{Z}})$,*

$$\theta \sim_{obs} \hat{\theta} \implies \theta \sim_{set} \hat{\theta}.$$

### 3.2 IMPLICATIONS OF GENERALIZATION

In the previous section, we introduced a new characterization of identifiability suited to general, unconstrained settings. Built from basic set-theoretic operations, this formulation appears flexible and composable. Yet it remains unclear how general it truly is, and more importantly, why that generality matters. To answer this, we examine its implications through a concrete example in Fig. 3, highlighting both its expressive power and practical utility. We begin with several interesting implications of the generalization:

**Proposition 1** (Implications of generalized identifiability). *For any two models $\theta = (g, p_Z)$ and $\hat{\theta} = (\hat{g}, p_{\hat{Z}})$, if $\theta \sim_{set} \hat{\theta}$, then for any two sets of observed variables $X_K$ and $X_V$, and their corresponding latent index sets $I_K$ and $I_V$, $Z_i$ is not a function of $\hat{Z}_{\pi(j)}$ for all $(i, j)$ satisfying at least one of the following, where $\pi$ is a permutation:*

*(i) (Object-centric) $i \in I_K$, $j \in I_V \setminus I_K$ or $i \in I_V$, $j \in I_K \setminus I_V$;*

*(ii) (Individual-centric) $i \in (I_K \setminus I_V)$, $j \in I_V$, or $i \in (I_V \setminus I_K)$, $j \in I_K$;*

*(iii) (Shared-centric) $i \in I_K \cap I_V$, $j \in I_K \Delta I_V$.*

**Example 3.** *In Fig. 3, Prop. 1 implies that, if we consider two groups of observed variables, such as $X1$ and $\{X_2, X_3\}$, regions like $I_1 \setminus (I_2 \cup I_3)$ illustrate individual-centric disentanglement, where latents unique to one group must be disentangled from the rest. Regions such as $I_1$, $I_2$, or $I_3$ represent object-centric disentanglement, where latents relevant to a single object must remain disentangled from the rest. The shared part can also be disentangled in a similar manner.*

**Why does it matter in the real world?** These implication types correspond to meaningful structures in real-world tasks. Object-centric disentanglement aligns with modularity in object-centric learning, where each object should have its own latent representation. Individual-centric disentanglement supports domain adaptation by isolating domain-specific factors. Shared-centric disentanglement captures common factors across domains or entities, which is essential for transferability and generalization. These patterns emerge naturally from set-theoretic indeterminacy and offer a principled way to design models that reflect the genus-differentia structure.

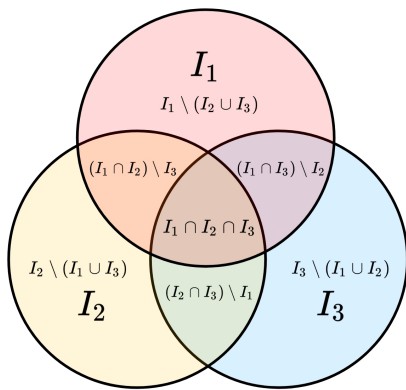

Figure 3: Running example.

**Atomic regions in the Venn diagram.** If the union of the latent index sets covers the full latent space, the generalized identifiability guarantees in Defn. 5 extend to every atomic region in the corresponding Venn diagram.[3]

**Example 4** (Identifying atomic regions). *Let $I_1$, $I_2$, and $I_3$ be the latent index sets associated with three observed variables in Fig. 3. Consider the atomic region $(I_1 \cap I_2) \setminus I_3$. To disentangle it from the rest, we first take $X_K = X_1$, $X_V = X_2$, so $I_K = I_1$, $I_V = I_2$. Then $i \in (I_1 \cap I_2) \setminus I_3 \subseteq I_K \cap I_V$, and $j \in I_K \Delta I_V = (I_1 \setminus I_2) \cup (I_2 \setminus I_1)$, ensuring $Z_i$ is not a function of $Z_j$ in the symmetric difference. Second, take $X_K = X_1 \cup X_2$, $X_V = X_3$, so $I_K = I_1 \cup I_2$ and $I_V = I_3$. Then $i \in (I_1 \cap I_2) \setminus I_3 \subseteq I_K \setminus I_V$ and $j \in I_3 = I_V$, ensuring $Z_i$ is also disentangled from latents of $X_3$. Together, these guarantee that the atomic region $(I_1 \cap I_2) \setminus I_3$ is disentangled from the rest. Other cases follow similarly and are in Appx. B. Therefore, each atomic region is disentangled from all other variables, and thus is region-wise (block-wise) identifiable[4] under the invertibility condition.*

---

[3]An atomic region in the Venn diagram is a non-empty set of the form $\bigcap_{i=1}^{n} B_i$, where each $B_i \in \{I_i, [d_z] \setminus I_i\}$ for a finite collection of sets $\{I_1, \ldots, I_n\}$. In the example, these correspond to all 7 distinct regions.

[4]A model is block-wise identifiable (Von Kügelgen et al., 2021) if the mapping between the estimated and ground-truth latent variables is a composition of block-wise invertible functions and permutations. Intuitively, variables can be entangled within the same block (set) but not across different blocks (sets).

**Remark 2** (Connection to block-identifiability). *By leveraging basic set-theoretic operations, we can construct a Venn diagram over the latent supports and identify all atomic regions, each defined as a minimal, non-overlapping region closed under finite intersections and complements. This perspective is conceptually related to block-wise identifiability (Von Kügelgen et al., 2021; Li et al., 2023; Yao et al., 2024b), but differs fundamentally in its assumptions and goals. Prior work achieves block identifiability by exploiting additional information such as multiple views or domains (Von Kügelgen et al., 2021; Yao et al., 2024b; Li et al., 2023). In contrast, our approach requires no such weak supervision. Notably, Yao et al. (2024b) also proposes an identifiability algebra, but in the opposite direction: they show that the intersection of latent groups can be identified after the groups themselves are recovered using multi-view signals. Our formulation instead starts from basic assumptions without any additional information, and directly targets the identifiability of intersections and complements without relying on external (weak) supervision. Of course, since the goals and setups are fundamentally different, our results do not supersede existing block-identifiability results, but rather offer a complementary perspective on recovering local structures.*

### 3.3 GENERALIZED IDENTIFIABILITY

Having established the characterization and implications of generalized identifiability, we now turn to its formal proof. Let $H$ denote a matrix sharing the support of the matrix-valued function $h$ in the identity $D_Z g(z) h(z, \hat{z}) = D_{\hat{Z}} \hat{g}(\hat{z})$. We begin by introducing the following assumption, which ensures sufficient nonlinearity in the system. Some arguments are omitted for brevity.

**Assumption 1** (**Sufficient nonlinearity**). *For each $i \in [d_x]$, there exists a set $S_i$ of $\|(D_Z g)_{i,\cdot}\|_0$ points such that the corresponding vectors for a model $(g, p_Z)$:*

$$\left( \frac{\partial X_i}{\partial Z_1}, \frac{\partial X_i}{\partial Z_2}, \ldots, \frac{\partial X_i}{\partial Z_{d_z}} \right) \Big|_{z=z^{(k)}}, k \in S_i,$$

*are linearly independent, where $z^{(k)}$ denotes a sample with index $k$ and $\mathrm{supp}((D_Z g(z^{(k)})H)_{i,\cdot}) \subseteq \mathrm{supp}((D_{\hat{Z}} \hat{g})_{i,\cdot})$.*

**Interpretation.** The assumption ensures the connection between the structure and the nonlinear function. In the asymptotic cases, we can usually find several samples in which the corresponding Jacobian vectors are linearly independent, i.e., span the support space. The assumption of non-exceeding support at these points generally rules out a global cancellation of a genuine dependency across all samples, which is intuitively similar to a violation of faithfulness.

**Connection to the literature.** The sufficient nonlinearity assumption makes it feasible to draw the connection between the Jacobian and the structure. It has been widely used in the literature (Lachapelle et al., 2022; Zheng et al., 2022; Kong et al., 2023; Yan et al., 2023), and aligns with the sufficient variability assumption (Hyvärinen & Morioka, 2016; Khemakhem et al., 2020; Sorrenson et al., 2020; Lachapelle et al., 2022; Zhang et al., 2024; Lachapelle et al., 2024). While most prior works often focus on variability across environments, sufficient nonlinearity imposes variability in the Jacobians across multiple samples to span the support space, following the spirit in (Lachapelle et al., 2022; Zheng et al., 2022; Lachapelle et al., 2024).

Then we are ready to present our main theorem for the generalized identifiability:

**Theorem 1** (**Generalized identifiability**). *Consider any two models $\theta = (g, p_Z)$ and $\hat{\theta} = (\hat{g}, p_{\hat{Z}})$ following the process in Sec. 2. Suppose Assum. 1 holds and:*

    *i. The probability density of $Z$ is positive in $\mathbb{R}^{d_z}$;*

    *ii. (Sparsity regularization[5]) $\|D_{\hat{Z}} \hat{g}\|_0 \leq \|D_Z g\|_0$.*

*Then if $\theta \sim_{obs} \hat{\theta}$, we have generalized identifiability (Defn. 6), i.e., $\theta \sim_{set} \hat{\theta}$.*

We further show that the dependency structure is identifiable up to a standard relabeling indeterminacy, providing structural insight when the underlying connections are of interest.

**Theorem 2** (**Structure identifiability**). *Consider any two models $\theta = (g, p_Z)$ and $\hat{\theta} = (\hat{g}, p_{\hat{Z}})$ following the process in Sec. 2. Suppose assumptions in Thm. 1 hold. If $\theta \sim_{obs} \hat{\theta}$, the support of the Jacobian matrix $D_{\hat{z}} \hat{g}$ is identical to that of $D_z g$, up to a permutation of column indices.*

---

[5]Notably, this is a regularization during estimation, instead of an assumption restricting the data.

**Universal inductive bias.** The first additional condition of positive density is standard and appears in nearly all previous identifiability results. We therefore focus on the sparsity regularization. Note that this is *not an assumption* on the data-generating process itself, but a practical inductive bias applied only during estimation. Thus, **the ground-truth process does *not* need to be sparse at all**.

This dependency sparsity reflects an inductive bias toward the simplicity of the hidden world. Among the many interpretations of Occam's razor, our approach aligns with the connectionist view, which prefers to always shave away unnecessary relations. This principle is fundamental and has been extensively studied in several fields. For example, in structural causal models, fully connected graphs are always Markovian to the observed distribution, but principles such as faithfulness, frugality, and minimality are used to eliminate spurious or redundant edges, revealing the true causal structure (Zhang, 2013). These simplicity criteria have been validated both theoretically and empirically over decades, supporting the use of sparsity as a reasonable inductive bias during regularization. Moreover, this regularization is highly practical: it can be integrated into most differentiable models, as long as gradients of the mappings with respect to latent variables are accessible.

### 3.4 FROM SETS TO ELEMENTS

Having established generalized identifiability through set-theoretic indeterminacy, which provides meaningful guarantees when full recovery is out of reach, a natural question arises: can stronger results, such as element identifiability for all latent variables, as targeted by most prior work, be obtained by imposing additional constraints?

**Definition 7** (**Element-wise indeterminacy**). *We say there is an **element-wise indeterminacy** between two models $\theta = (g, p_Z)$ and $\hat{\theta} = (\hat{g}, p_{\hat{Z}})$, denoted $\theta \sim_{elem} \hat{\theta}$, if and only if*

$$\hat{Z} = P_\pi \varphi(Z),$$

*where $\varphi(Z) = (\varphi_1(Z_1), \ldots, \varphi_{d_z}(Z_{d_z})), \varphi : \mathcal{Z} \implies \hat{\mathcal{Z}}$ is a element-wise diffeomorphism and $P_\pi$ is a permutation matrix corresponding to a $d_z$-permutation $\pi$.*

**Definition 8** (**Element identifiability** (Hyvärinen & Pajunen, 1999)). *For a model $\theta = (g, p_Z)$, we have **element identifiability**, if and only if, for any other model $\hat{\theta} = (\hat{g}, p_{\hat{Z}})$,*

$$\theta \sim_{obs} \hat{\theta} \implies \theta \sim_{elem} \hat{\theta}.$$

**Connection to generalized identifiability.** Generalized identifiability (Defn. 6) focuses on recovering partial information from subsets of observed variables, while permutation identifiability seeks to recover all latent variables up to element-wise indeterminacy, which is strictly stronger. As a result, achieving permutation identifiability naturally requires stronger assumptions.

Interestingly, this can be a natural consequence of set-theoretic indeterminacy. As discussed in Sec. 3.2, generalized identifiability guarantees the recovery of atomic regions in the Venn diagram. Therefore, if the Venn diagram is sufficiently rich, meaning that each latent variable corresponds to its own atomic region, we obtain element-wise identifiability directly. Since the Venn diagram is simply a representation of the dependency structure, we now formalize the corresponding structural condition as follows. For each $X_j \in A$, let $I_j$ be the index set of latent variables connected to $X_j$.

**Assumption 2** (**Sufficient diversity**). *For each latent variable $Z_i$ ($i \in [d_z]$), there exists a set of observed variables $A$ such that* at least one *of the following three conditions holds:*

1. *There exists $X_k \in A$ such that $\bigcup_{X_j \in A} I_j = [d_z], I_k \setminus \bigcup_{X_j \in A \setminus \{X_k\}} I_j = i$.*

2. *There exists $X_k \in A$ such that $\bigcup_{X_j \in A} I_j = [d_z], \left( \bigcap_{X_j \in A \setminus \{X_k\}} I_j \right) \setminus I_k = i$.*

3. *(Zheng et al., 2022) The intersection of supports satisfies $\bigcap_{X_j \in A} I_j = i$.*

**Theorem 3** (**Element identifiability**). *Consider any two models $\theta = (g, p_Z)$ and $\hat{\theta} = (\hat{g}, p_{\hat{Z}})$ following the process in Sec. 2. Suppose assumptions in Thm. 1 and Assum. 2 hold. Then we have identifiability up to element-wise indeterminacy, i.e., $\theta \sim_{obs} \hat{\theta} \implies \theta \sim_{elem} \hat{\theta}$.*

**Generalized structural condition.** The *sufficient diversity* serves as a generalized condition for element identifiability in fully unsupervised settings. The most closely related work is Zheng et al. (2022), which also derives nonparametric identifiability results based purely on structural assumptions, without relying on auxiliary variables, interventions, or restrictive functional forms. However,

their structural sparsity condition aligns exactly with the third clause of *sufficient diversity*, making it strictly stronger. In contrast, our formulation introduces two additional conditions as *alternatives*, expanding the class of admissible structures and offering greater flexibility. We conjecture that sufficient diversity may even be necessary when no distributional or functional form constraints are imposed, as it arises naturally from the structure of atomic regions in the Venn diagram. Since any dependency structure admits such a representation, and atomic regions serve as its minimal elements, our condition may capture the essential structural requirement for element-level recovery.

**Diversity is not sparsity.** It is worth emphasizing that our diversity condition is fundamentally different from sparsity assumptions. Diversity does not require the structure to be sparse: it remains valid even in nearly fully connected settings, as long as there is some variation (e.g., even a single differing edge) in the connectivity patterns across variables. By contrast, sparsity-based assumptions strictly enforce sparse structures. For example, the well-known anchor feature assumption (Arora et al., 2012; Moran et al., 2021) requires each latent variable to have at least two observed variables that are unique to it, thereby excluding dense structures.

## 4 EXPERIMENT

In this section, we provide empirical support for our results in both synthetic and real-world settings. Due to page limits, **additional experimental results are deferred to Appendix C**, including (1) *comparisons of more Jacobian/Hessian penalties* (e.g., (Wei et al., 2021; Peebles et al., 2020)), (2) *analyses of regularization weights*, and (3) *further visual results on synthetic and real data*.

### 4.1 SYNTHETIC EXPERIMENTS

**Setup.** We follow the data generation process in Sec. 2. We employ the variational autoencoder as our backbone model with a dependency sparsity regularization in the objective function as:

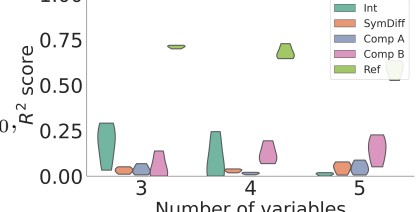

Figure 4: $R^2$ in simulation.

$$\mathcal{L} = \underbrace{\mathbb{E}_{q(Z|X)}[\ln p(X|Z)] - \beta D_{KL}(q(Z|X)\|p(Z))}_{\text{Evidence Lower Bound}} + \alpha\|D_{\hat{z}\hat{g}}\|_0,$$

where $D_{KL}$ is the Kullback–Leibler divergence, $q(Z|X)$ the variational posterior, $p(Z)$ the prior, $p(X|Z)$ the likelihood, and $\alpha, \beta$ regularization weights. We use $10,000$ samples and set $\alpha = \beta = 0.05$ for all experiments, and all generation processes are nonlinear, implemented by MLPs with Leaky ReLU.

**Generalized Identifiability.** We begin by evaluating generalized identifiability across groups of observed variables. We generate datasets with dimensionality in $\{3, 4, 5\}$ and split the observed variables into two groups, $X_K$ and $X_V$. For each dataset, we compute the $R^2$ score, lower means more disentangled, between: (1) *Int*, $I_K \cap I_V$ and $I_K \Delta I_V$; (2) *SymDiff*, $I_K \Delta I_V$ and $I_K \cap I_V$; and (3) *Comp A* and *Comp B*, both directions between $I_K \setminus I_V$ and $I_V \setminus I_K$. We also include *Ref*, the $R^2$ between $Z$ and $\hat{Z}$, as a baseline indicating the expected level of $R^2$ for entangled variables. As shown in Fig. 4, all disentan-

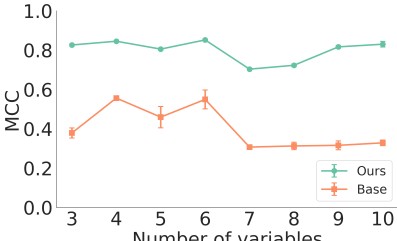

Figure 5: $MCC$ in simulation.

glement conditions implied by set-theoretic indeterminacy (Defn. 5) are satisfied: the $R^2$ between structurally disjoint components is consistently much lower than *Ref*, supporting the validity of generalized identifiability.

**Element Identifiability.** We evaluate whether comparing multiple variable pairs enables recovery of latent variables up to element-wise indeterminacy. We construct datasets with varying dimensions and structures that either satisfy Sufficient Diversity (Assum. 2) (*Ours*) or violate it through fully dense dependencies (*Base*). Following prior work (Hyvärinen et al., 2024), we use the mean correlation coefficient (MCC) between estimated and ground-truth latent variables as the evaluation metric. As shown in Fig. 5, only datasets satisfying the structural condition achieve high MCC, confirming that element-wise identifiability holds under our assumptions.

| Method | Shapes3D | | Cars3D | | MPI3D | |
|---|---|---|---|---|---|---|
| | FactorVAE ↑ | DCI ↑ | FactorVAE ↑ | DCI ↑ | FactorVAE ↑ | DCI ↑ |
| *VAE-based* | | | | | | |
| FactorVAE (Kim & Mnih, 2018) | $0.833 \pm 0.025$ | $0.484 \pm 0.120$ | $0.708 \pm 0.026$ | $0.135 \pm 0.030$ | $0.599 \pm 0.064$ | $0.345 \pm 0.047$ |
| FactorVAE + Latent Sparsity | $0.837 \pm 0.069$ | $0.477 \pm 0.152$ | $0.501 \pm 0.434$ | $0.113 \pm 0.069$ | $0.440 \pm 0.065$ | $0.325 \pm 0.028$ |
| FactorVAE + Dependency Sparsity | $\mathbf{0.871 \pm 0.053}$ | $\mathbf{0.575 \pm 0.032}$ | $\mathbf{0.752 \pm 0.040}$ | $\mathbf{0.144 \pm 0.053}$ | $\mathbf{0.639 \pm 0.084}$ | $\mathbf{0.384 \pm 0.031}$ |
| *Diffusion-based* | | | | | | |
| EncDiff (Yang et al., 2024) | $0.9999 \pm 0.0001$ | $0.901 \pm 0.050$ | $\mathbf{0.779 \pm 0.060}$ | $0.250 \pm 0.020$ | $0.868 \pm 0.033$ | $0.676 \pm 0.018$ |
| EncDiff + Latent Sparsity | $0.967 \pm 0.042$ | $0.891 \pm 0.057$ | $0.729 \pm 0.003$ | $0.241 \pm 0.016$ | $0.879 \pm 0.015$ | $\mathbf{0.684 \pm 0.020}$ |
| EncDiff + Dependency Sparsity | $\mathbf{1.0000 \pm 0.0000}$ | $\mathbf{0.947 \pm 0.005}$ | $0.756 \pm 0.041$ | $\mathbf{0.256 \pm 0.011}$ | $\mathbf{0.881 \pm 0.024}$ | $0.667 \pm 0.047$ |
| *GAN-based* | | | | | | |
| DisCo (Ren et al., 2021) | $0.852 \pm 0.037$ | $0.710 \pm 0.020$ | $0.727 \pm 0.106$ | $0.319 \pm 0.031$ | $0.396 \pm 0.023$ | $0.306 \pm 0.079$ |
| DisCo + Latent Sparsity | $0.864 \pm 0.007$ | $0.707 \pm 0.024$ | $0.761 \pm 0.148$ | $0.294 \pm 0.023$ | $0.308 \pm 0.031$ | $0.314 \pm 0.050$ |
| DisCo + Dependency Sparsity | $\mathbf{0.868 \pm 0.017}$ | $\mathbf{0.712 \pm 0.018}$ | $\mathbf{0.789 \pm 0.029}$ | $\mathbf{0.320 \pm 0.003}$ | $\mathbf{0.410 \pm 0.122}$ | $\mathbf{0.324 \pm 0.059}$ |

Table 1: Comparison of disentanglement on FactorVAE score and DCI (mean±std, higher is better). Bold numbers denote the best value *within each model family* for a given dataset/metric.

## 4.2 VISUAL EXPERIMENTS

**Setup.** Following the literature, we evaluate identification in more complex settings by learning latent variables as generative factors. Specifically, we follow the setting of (Yang et al., 2024) and use three standard benchmark datasets of disentangled representation learning: Cars3D (Reed et al., 2015), Shapes3D (Kim & Mnih, 2018), and MPI3D (Gondal et al., 2019), which are benchmark datasets with known generative factors such as object color, shape, scale, orientation, and viewpoint, ranging from synthetic renderings to real-world images.

To evaluate the effectiveness of the proposed sparsity loss, we incorporate it into three powerful disentangled representation learning methods based on mainstream generative models: Variational Autoencoders (VAE), Generative Adversarial Networks (GAN), and Diffusion Models. These methods correspond to FactorVAE (Kim & Mnih, 2018), DisCo (Ren et al., 2021), and EncDiff (Yang et al., 2024), respectively. We consider two types of baselines: 1) the original methods, i.e., FactorVAE, DisCo, and EncDiff, and 2) versions of these methods that incorporate L1 regularization on $Z$ (latent sparsity). In contrast, our approach applies an L1 regularization on Jacobian (dependency sparsity). Following standard practice, we use FactorVAE score (Kim & Mnih, 2018) and the DCI Disentanglement score (Eastwood & Williams, 2018) as evaluation metrics. We repeat each method over three random seeds. Please refer to Appx. C for more details on setups.

**Dependency sparsity in the literature.** Notably, dependency sparsity has been widely used as a simple and standard regularization across diverse settings, from disentanglement to LLMs (Rhodes & Lee, 2021; Zheng et al., 2022; Farnik et al., 2025), although a general identifiability theory is still lacking. Thus, its empirical effectiveness is already well established, and our experiments aim to provide further supporting evidence.

**Latent or dependency sparsity?** Table 1 shows that across most datasets and backbone methods, introducing the proposed dependency sparsity consistently helps the understanding of the hidden world. Notably, these generative models often benefit more from dependency sparsity than from latent sparsity. This is particularly interesting given the widespread use of sparse latent regularization in mechanistic interpretability (e.g., sparse autoencoders (Cunningham et al., 2023)). Our results highlight not only the advantage of dependency sparsity, but also lend insight to recent concerns about the limitations of latent sparsity raised in the interpretability literature, such as feature absorption, linear constraints, and high dimensionality (Sharkey et al., 2025).

## 5 CONCLUSION

We introduce *diverse dictionary learning* to investigate which aspects of the hidden world can be recovered under basic conditions, and which inductive biases may be universally beneficial during estimation. Our guarantees, grounded in set algebra, offer a complementary local view to prior results based on global assumptions, and also unify existing structural conditions for full identifiability. For future work, it is worth exploring generalized identifiability in foundation models. Current models are largely driven by empirical insights, and inductive biases inspired by identifiability, which have been overlooked, may offer fresh directions for breakthroughs. With massive data and computation available, asymptotic guarantees are becoming increasingly relevant, making identifiability practically significant. A deeper investigation along this line remains an open limitation of our work.

## ACKNOWLEDGMENT

The authors would like to thank the anonymous reviewers, AC, Lucas Jr, and Alan Amin for helpful comments and suggestions. The authors would also like to acknowledge the support from NSF Award No. 2229881, AI Institute for Societal Decision Making (AI-SDM), the National Institutes of Health (NIH) under Contract R01HL159805, and grants from Quris AI, Florin Court Capital, MBZUAI-WIS Joint Program, and the Al Deira Causal Education project.

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

# Diverse Dictionary Learning
**Supplementary Material**

## Table of Contents

| Symbol | Description |
|--------|-------------|
| $X = (X_1, \ldots, X_{d_x}) \in \mathbb{R}^{d_x}$ | Observed variables (data space) |
| $Z = (Z_1, \ldots, Z_{d_z}) \in \mathbb{R}^{d_z}$ | Latent variables (hidden space) |
| $g : \mathbb{R}^{d_z} \to \mathbb{R}^{d_x}$ | Generative map, diffeomorphism onto its image |
| $\text{supp}(M; \Theta)$ | Support of a matrix-valued function $M : \Theta \to \mathbb{R}^{m \times n}$ |
| $S = \text{supp}(D_z g; \mathcal{Z})$ | Dependency structure: support of Jacobian between $Z$ and $X$ |
| $\theta = (g, p_Z)$ | Model consisting of generative map and latent distribution |
| $\theta \sim_{\text{obs}} \hat{\theta}$ | Observational equivalence (same induced distribution on $X$) |
| $\theta \sim_{\text{set}} \hat{\theta}$ | Set-theoretic indeterminacy (intersection, symmetric difference, complement disentangled) |
| $I_S \subseteq [d_z]$ | Latent index set associated with observed variable set $X_S$ |
| $I_K \cap I_V$ | Intersection of latent supports (shared factors) |
| $I_K \Delta I_V$ | Symmetric difference of latent supports (unique factors) |
| $I_K \setminus I_V, \ I_V \setminus I_K$ | Complements (exclusive latent components) |
| Atomic region | Minimal block in Venn diagram defined by intersections and complements of latent supports |
| $\theta \sim_{\text{elem}} \hat{\theta}$ | Element-wise indeterminacy (permutation + invertible reparametrization) |

Table 2: Notation used throughout the paper.

# A PROOFS

## A.1 PROOF OF PROPOSITION 1

**Proposition 1** (Implications of generalized identifiability). *For any two models $\theta = (g, p_Z)$ and $\hat{\theta} = (\hat{g}, p_{\hat{Z}})$, if $\theta \sim_{set} \hat{\theta}$, then for any two sets of observed variables $X_K$ and $X_V$, and their corresponding latent index sets $I_K$ and $I_V$, $Z_i$ is not a function of $\hat{Z}_{\pi(j)}$ for all $(i, j)$ satisfying at least one of the following, where $\pi$ is a permutation:*

- *(i) (Object-centric) $i \in I_K, j \in I_V \setminus I_K$ or $i \in I_V, j \in I_K \setminus I_V$;*

- *(ii) (Individual-centric) $i \in (I_K \setminus I_V), j \in I_V$, or $i \in (I_V \setminus I_K), j \in I_K$;*

- *(iii) (Shared-centric) $i \in I_K \cap I_V, j \in I_K \Delta I_V$.*

*Proof.* For $\theta = (g, p_Z)$ and $\hat{\theta} = (\hat{g}, p_{\hat{Z}})$, since $\theta \sim_{\text{set}} \hat{\theta}$, for any two sets of observed variables $X_K$ and $X_V$, and their corresponding latent index sets $I_K$ and $I_V$, there exists a permutation $\pi$ over $\{1, \ldots, d_z\}$ such that $Z_i$ is not a function of $\hat{Z}_{\pi(j)}$ for any $(i, j)$ satisfying at least one of the following conditions:

- (i) (*Intersection*) $i \in I_K \cap I_V, j \in I_K \Delta I_V$;

- (ii) (*Symmetric difference*) $i \in I_K \Delta I_V, j \in I_K \cap I_V$;

- (iii) (*Complement*) $i \in I_K \setminus I_V, j \in I_V \setminus I_K$, or $i \in I_V \setminus I_K, j \in I_K \setminus I_V$.

Our goal is to prove that, the same holds for all $(i, j)$ satisfying at least one of the following conditions:

- (i) (*Object-centric disentanglement*) $i \in I_K, j \in I_V \setminus I_K$ or $i \in I_V, j \in I_K \setminus I_V$;

- (ii) (*Individual-centric disentanglement*) $i \in I_K \setminus I_V, j \in I_V$, or $i \in I_V \setminus I_K, j \in I_K$;

- (iii) (*Shared-centric disentanglement*) $i \in I_K \cap I_V, j \in I_K \Delta I_V$.

Let us start with the first case. If $i \in I_K$, it is either the case $i \in I_K \cap I_V$ or $i \in I_K \setminus I_V$. For $i \in I_K \cap I_V$, according to the case of *Intersection* in set-theoretic indeterminacy, we have

$$\frac{\partial Z_i}{\partial \hat{Z}_{\pi(j)}} = 0, \tag{2}$$

for any $j \in I_K \Delta I_V$. Similarly, for $i \in I_K \setminus I_V$, we also have Eq. (2) for $j \in I_V \setminus I_K$. Combining these together, Eq. (2) must holds for any $i \in I_K$ and $j \in I_V \setminus I_K$. The similar derivation holds for any $i \in I_V$ and $j \in I_K \setminus I_V$. Thus, the first case holds.

Then we consider the second case. If $i \in I_K \setminus I_V$, then according to the case of *symmetric difference* in set-theoretic indeterminacy, we have Eq. (2) holds for $j \in I_K \cap I_V$.

Moreover, according to the case of *complement* in set-theoretic indeterminacy, we have Eq. (2) holds for $j \in I_V \setminus I_K$. Note that there is

$$(I_V \setminus I_K) \cup (I_K \cap I_V) = I_V. \tag{3}$$

Thus, for any $i \in I_K \setminus I_V$, we have Eq. (2) holds for any $j \in I_V$. The similar derivation holds for any $i \in I_V \setminus I_K$ and $j \in I_K$. Thus, the second case holds.

The third case is identical to the case of *intersection* in set-theoretic indeterminacy. Thus, for $\theta = (g, p_Z)$ and $\hat{\theta} = (\hat{g}, p_{\hat{Z}})$, $\theta \sim_{\text{set}} \hat{\theta}$ implies our goals. $\qquad \square$

## A.2 PROOF OF THEOREM 1

**Theorem 1 (Generalized identifiability).** *Consider any two models* $\theta = (g, p_Z)$ *and* $\hat{\theta} = (\hat{g}, p_{\hat{Z}})$ *following the process in Sec. 2. Suppose Assum. 1 holds and:*

  *i. The probability density of $Z$ is positive in $\mathbb{R}^{d_z}$;*

  *ii. (Sparsity regularization[6]) $\|D_{\hat{Z}}\hat{g}\|_0 \leq \|D_Z g\|_0$.*

*Then if $\theta \sim_{\text{obs}} \hat{\theta}$, we have generalized identifiability (Defn. 6), i.e., $\theta \sim_{\text{set}} \hat{\theta}$.*

*Proof.* Since $\theta \sim_{\text{obs}} \hat{\theta}$, by the change-of-variable formula there must be

$$\hat{Z} = \hat{g}^{-1} \circ g(Z) = \phi(Z), \tag{4}$$

where $\phi = \hat{g}^{-1} \circ g$ is an invertible function and thus $\phi^{-1}$ exists. Therefore, according to the chain rule, we have

$$D_{\hat{Z}}\hat{g} = D_Z g D_{\hat{Z}} \phi^{-1}. \tag{5}$$

For each $i \in [d_x]$, consider a set $S_i$ of $\|(D_Z g)_{i,\cdot}\|_0$ distinct points and the corresponding Jacobians as follows

$$\left( \frac{\partial X_i}{\partial Z_1}, \frac{\partial X_i}{\partial Z_2}, \ldots, \frac{\partial X_i}{\partial Z_{d_z}} \right) \Bigg|_{(z) = (z^{(k)})}, k \in S_i. \tag{6}$$

According to Assumption 1, all vectors in Eq. (6) are linearly independent.

Let us construct a matrix $M_\phi$. Since all vectors in Eq. (6) are linearly independent, for any $j \in \text{supp}((D_Z g)_{i,\cdot})$, we have

$$M_{\phi j,\cdot} = \sum_{k \in S_i} \beta_k (D_Z g(z^{(k)}))_{i,\cdot} M_\phi, \tag{7}$$

where $\beta_k, \forall k \in S_i$ denote coefficients, and $M_\phi$ denotes a matrix.

We wish to construct a constant matrix $M_\phi$ satisfying

$$\sum_{k \in S_i} \beta_k (D_Z g(z^{(k)}))_{i,\cdot} M_\phi \in \text{span}\{e_j : j \in \text{supp}((D_{\hat{Z}}\hat{g})_{i,\cdot})\}, \tag{8}$$

for each $i \in [d_x]$, while ensuring that

$$\text{supp}(M_\phi) = \text{supp}(D_{\hat{Z}} \phi^{-1}), \tag{9}$$

---

[6]Notably, this is a regularization during estimation, instead of an assumption restricting the data.

According to Assumption 1, we have

$$\text{supp}(D_Z g(z^{(k)}) M_\phi)_{i,\cdot} \subseteq \text{supp}(D_{\hat{Z}} \hat{g}(\hat{z}^{(k)}))_{i,\cdot}, \forall k \in S_i. \tag{10}$$

Therefore, there must be

$$D_Z g(z^{(k)}))_{i,\cdot} M_\phi \in \text{span}\{e_j : j \in \text{supp}((D_{\hat{Z}} \hat{g})_{i,\cdot})\}, \tag{11}$$

which implies

$$\sum_{k \in S_i} \beta_k \left(D_Z g(z^{(k)})\right)_{i,\cdot} M_\phi \in \text{span}\{e_j : j \in \text{supp}((D_{\hat{Z}} \hat{g})_{i,\cdot})\}. \tag{12}$$

Equivalently, we have

$$M_{\phi_{j,\cdot}} \in \text{span}\{e_k : k \in \text{supp}((D_{\hat{Z}} \hat{g})_{i,\cdot})\}, \forall j \in \text{supp}((D_Z g)_{i,\cdot}). \tag{13}$$

Define a bipartite graph $G = (R, C, E)$ where $R = C = \{1, 2, \ldots, d_z\}$ and an edge exists between $j \in R$ and $k \in C$ if and only if $D_{\hat{Z}} \phi_{j,k}^{-1} \neq 0$.

Since $D_{\hat{Z}} \phi^{-1}$ is invertible, its rows are linearly independent, so for every subset $S \subseteq R$, the corresponding rows have a nonzero determinant, implying that

$$|\{k \in C \mid \exists j \in S, D_{\hat{Z}} \phi_{j,k}^{-1} \neq 0\}| \geq |S|. \tag{14}$$

By Hall's marriage theorem, there exists a perfect matching between $R$ and $C$. This matching corresponds to a permutation $\pi \in S_n$ such that

$$(D_{\hat{Z}} \phi^{-1})_{j,\pi(j)} \neq 0, \forall j \in \{1, 2, \ldots, n\}. \tag{15}$$

In particular, for every $j \in \text{supp}((D_Z g)_{i,\cdot}) \subseteq \{1, 2, \ldots, n\}$, we have

$$(D_{\hat{Z}} \phi^{-1})_{j,\pi(j)} \neq 0. \tag{16}$$

Because $\text{supp}(M_\phi) = \text{supp}(D_{\hat{Z}} \phi^{-1})$, this implies

$$M_{\phi_{j,\pi(j)}} \neq 0, \forall j \in \text{supp}((D_Z g)_{i,\cdot}). \tag{17}$$

Further incorporating Eq. (13), it follows that

$$\pi(j) \in \text{span}\{e_k : k \in \text{supp}((D_{\hat{Z}} \hat{g})_{i,\cdot})\}, \forall j \in \text{supp}((D_Z g)_{i,\cdot}). \tag{18}$$

Therefore, for any non-zero element in $D_z g$, there always exists a corresponding non-zero element in $D_{\hat{z}} \hat{g}$, with the relations on their indices as follows

$$(D_z g)_{i,j} \neq 0 \implies (D_{\hat{z}} \hat{g})_{i,\pi(j)} \neq 0. \tag{19}$$

Furthermore, because of the assumption that

$$\|D_{\hat{Z}} \hat{g}\|_0 \leq \|D_Z g\|_0, \tag{20}$$

Eq. (19) can be further restricted to an equivalence between the sparsity patterns, i.e.,

$$(D_z g)_{i,j} \neq 0 \iff (D_{\hat{z}} \hat{g})_{i,\pi(j)} \neq 0 \tag{21}$$

We then consider the following two cases for the set-theoretic indeterminacy. Specifically, for any two sets of observed variables $X_K$ and $X_V$ and the index sets of their latent variables, $I_K$ and $I_V$, $K \neq V$, we consider the following cases:

(i) (*Intersection*) $i \in I_K \cap I_V, j \in I_K \Delta I_V$;

(ii) (*Symmetric difference*) $i \in I_K \Delta I_V, j \in I_K \cap I_V$;

(iii) (*Complement*) $i \in I_K \setminus I_V, j \in I_V \setminus I_K$, or $i \in I_V \setminus I_K, j \in I_K \setminus I_V$.

Let us start from the first case, where $t \in I_K \cap I_V$, $r \in I_K \Delta I_V$. Denote the index sets of $X_K$ and $X_V$ as $J_K$ and $J_V$. Then, there exists $k \in J_K$ such that

$$t \in \text{supp}(D_Z g)_{k,\cdot}. \tag{22}$$

This further implies the following relation based on Eq. (13)

$$M_{\phi_{t,\cdot}} \in \text{span}\{e_{k'} : k' \in \text{supp}((D_{\hat{Z}} \hat{g})_{k,\cdot})\}. \tag{23}$$

Similarly, there exists $v \in J_V$ such that

$$t \in \text{supp}(D_Z g)_{v,\cdot}, \tag{24}$$

which further implies

$$M_{\phi_{t,\cdot}} \in \text{span}\{e'_k : k' \in \text{supp}((D_{\hat{Z}} \hat{g})_{v,\cdot})\}. \tag{25}$$

For $r \in I_K \Delta I_V$, suppose

$$M_{\phi_{t,\pi(r)}} \neq 0. \tag{26}$$

According to Eqs. (23) and (25), there must be

$$\pi(r) \in \text{supp}(D_{\hat{Z}} \hat{g})_{k,\cdot}, \tag{27}$$

$$\pi(r) \in \text{supp}(D_{\hat{Z}} \hat{g})_{v,\cdot}. \tag{28}$$

Together with Eq. (21), these further imply

$$r \in \text{supp}(D_Z g)_{k,\cdot}, \tag{29}$$

$$r \in \text{supp}(D_Z g)_{v,\cdot}. \tag{30}$$

This leads to

$$r \in I_K \cap I_V, \tag{31}$$

which contradict $r \in I_K \Delta I_V$. Therefore, there must be

$$M_{\phi_{t,\pi(r)}} = 0. \tag{32}$$

Since $M_\phi$ is the support of $D_{\hat{Z}} \phi^{-1}$, this implies that, for $t \in I_K \cap I_V$ and $r \in I_K \Delta I_V$, we have

$$\frac{\partial Z_t}{\partial \hat{Z}_{\pi(r)}} = 0. \tag{33}$$

Then we consider the case where $t \in I_K \cap I_V$ and $r \in I \setminus (I_K \cup I_V)$. Suppose

$$M_{\phi_{t,\pi(r)}} \neq 0. \tag{34}$$

According to Eq. (23), there must be

$$\pi(r) \in \text{supp}(D_{\hat{Z}} \hat{g})_{k,\cdot}. \tag{35}$$

Together with Eq. (21), these further imply

$$r \in \text{supp}(D_Z g)_{k,\cdot}. \tag{36}$$

This leads to

$$r \in I_K, \tag{37}$$

which contradict $r \in I \setminus (I_K \cup I_V)$. Therefore, there must be

$$M_{\phi_{t,\pi(r)}} = 0, \tag{38}$$

where $t \in I_K \cap I_V$ and $r \in I \setminus (I_K \cup I_V)$.

Therefore, according to Eqs. (33) and (38) and the invertibility of $\phi$, for $t \in I_K \cap I_V$, $Z_t$ has to depend only on $\hat{Z}_{\pi(t)}$ and not other variables. Therefore, there exists an invertible function $h$ s.t. $Z_t = h(\hat{Z}_{\pi(t)})$.

Further consider the setting where $r \in I_K \Delta I_V$. Since $t \in I_K \cap I_V$ and $(I_K \Delta I_V) \cap (I_K \cap I_V) = \emptyset$, $Z_r$ is independent of $Z_t = h(\hat{Z}_{\pi(t)})$. Therefore, $Z_r$ does not depend on $\hat{Z}_{\pi(t)}$ and thus

$$\frac{\partial Z_r}{\partial \hat{Z}_{\pi(t)}} = 0, \tag{39}$$

which is the second case.

Then we consider the third case where $t \in I_K \setminus I_V$ and $r \in I_V \setminus I_K$. If $t \in I_K \setminus I_V$, there exists $k \in J_K$ such that

$$t \in \text{supp}(D_Z g)_{k,\cdot} \tag{40}$$

Then there is

$$M_{\phi_{t,\cdot}} \in \text{span}\{e_{k'} : k' \in \text{supp}((D_{\hat{Z}} \hat{g})_{k,\cdot})\}. \tag{41}$$

For $r \in I_V \setminus I_K$, suppose

$$M_{\phi_{t,\pi(r)}} \neq 0. \tag{42}$$

Then we have

$$\pi(r) \in \text{supp}(D_{\hat{Z}} \hat{g})_{k,\cdot}, \tag{43}$$

which follows

$$r \in \text{supp}(D_Z g)_{k,\cdot} \tag{44}$$

This is a contradiction since $r \in I_V \setminus I_K$. Thus, we can also prove that, for the third case, where $t \in I_V \setminus I_K$ and $r \in I_K \setminus I_V$, there must be

$$\frac{\partial Z_t}{\partial \hat{Z}_{\pi(r)}} = 0. \tag{45}$$

This concludes the proof.

$\square$

## A.3 PROOF OF THEOREM 2

**Theorem 2 (Structure identifiability).** *Consider any two models $\theta = (g, p_Z)$ and $\hat{\theta} = (\hat{g}, p_{\hat{Z}})$ following the process in Sec. 2. Suppose assumptions in Thm. 1 hold. If $\theta \sim_{obs} \hat{\theta}$, the support of the Jacobian matrix $D_{\hat{z}} \hat{g}$ is identical to that of $D_z g$, up to a permutation of column indices.*

*Proof.* Since $\theta \sim_{\text{obs}} \hat{\theta}$, by considering $\phi = \hat{g}^{-1} \circ g$ and the change-of-variable formula, we have

$$\hat{Z} = \phi(Z), \tag{46}$$

where $\phi$ is an invertible function and thus $\phi^{-1}$ exists. Therefore, according to the chain rule, we have

$$D_{\hat{Z}} \hat{g} = D_Z g D_{\hat{Z}} \phi^{-1}. \tag{47}$$

For each $i \in [d_x]$, consider a set $S_i$ of $\|(D_Z g)_{i,\cdot}\|_0$ distinct points and the corresponding Jacobians as follows

$$\left( \frac{\partial X_i}{\partial Z_1}, \frac{\partial X_i}{\partial Z_2}, \ldots, \frac{\partial X_i}{\partial Z_{d_z}} \right) \Bigg|_{(z)=(z^{(k)})}, k \in S_i. \tag{48}$$

According to Assumption 1, all vectors in Eq. (48) are linearly independent.

Let us construct a matrix $M_\phi$. Since all vectors in Eq. (48) are linearly independent, for any $j \in \text{supp}((D_Z g)_{i,\cdot})$, we have

$$M_{\phi_{j,\cdot}} = \sum_{k \in S_i} \beta_k (D_Z g(z^{(k)}))_{i,\cdot} M_\phi, \tag{49}$$

where $\beta_k, \forall k \in S_i$ denote coefficients.

We wish to construct a constant matrix $M_\phi$ satisfying

$$\sum_{k \in S_i} \beta_k (D_Z g(z^{(k)}))_{i,\cdot} M_\phi \in \text{span}\{e_j : j \in \text{supp}((D_{\hat{Z}} \hat{g})_{i,\cdot})\}, \tag{50}$$

for each $i \in [d_x]$, while ensuring that

$$\text{supp}(M_\phi) = \text{supp}(D_{\hat{Z}}\phi^{-1}), \tag{51}$$

According to Assumption 1, we have

$$\text{supp}(D_Z g(z^{(k)}) M_\phi)_{i,\cdot} \subseteq \text{supp}(D_{\hat{Z}}\hat{g}(\hat{z}^{(k)}))_{i,\cdot}, \forall k \in S_i. \tag{52}$$

Therefore, there must be

$$D_Z g(z^{(k)}))_{i,\cdot} M_\phi \in \text{span}\{e_j : j \in \text{supp}((D_{\hat{Z}}\hat{g})_{i,\cdot})\}, \tag{53}$$

which implies

$$\sum_{k \in S_i} \beta_k \left(D_Z g(z^{(k)})\right)_{i,\cdot} M_\phi \in \text{span}\{e_j : j \in \text{supp}((D_{\hat{Z}}\hat{g})_{i,\cdot})\}. \tag{54}$$

Equivalently, we have

$$M_{\phi_{j,\cdot}} \in \text{span}\{e_k : k \in \text{supp}((D_{\hat{Z}}\hat{g})_{i,\cdot})\}, \forall j \in \text{supp}((D_Z g)_{i,\cdot}). \tag{55}$$

Define a bipartite graph $G = (R, C, E)$ where $R = C = \{1, 2, \ldots, d_z\}$ and an edge exists between $j \in R$ and $k \in C$ if and only if $D_{\hat{Z}}\phi^{-1}_{j,k} \neq 0$.

Since $D_{\hat{Z}}\phi^{-1}$ is invertible, its rows are linearly independent, so for every subset $S \subseteq R$, the corresponding rows have a nonzero determinant, implying that

$$|\{k \in C \mid \exists j \in S, D_{\hat{Z}}\phi^{-1}_{j,k} \neq 0\}| \geq |S|. \tag{56}$$

By Hall's marriage theorem, there exists a perfect matching between $R$ and $C$. This matching corresponds to a permutation $\pi \in S_n$ such that

$$(D_{\hat{Z}}\phi^{-1})_{j,\pi(j)} \neq 0, \forall j \in \{1, 2, \ldots, n\}. \tag{57}$$

In particular, for every $j \in \text{supp}((D_Z g)_{i,\cdot}) \subseteq \{1, 2, \ldots, n\}$, we have

$$(D_{\hat{Z}}\phi^{-1})_{j,\pi(j)} \neq 0. \tag{58}$$

Because $\text{supp}(M_\phi) = \text{supp}(D_{\hat{Z}}\phi^{-1})$, this implies

$$M_{\phi_{j,\pi(j)}} \neq 0, \forall j \in \text{supp}((D_Z g)_{i,\cdot}). \tag{59}$$

Further incorporating Eq. (55), it follows that

$$\pi(j) \in \text{span}\{e_k : k \in \text{supp}((D_{\hat{Z}}\hat{g})_{i,\cdot})\}, \forall j \in \text{supp}((D_Z g)_{i,\cdot}). \tag{60}$$

Therefore, for any non-zero element in $D_z g$, there always exists a corresponding non-zero element in $D_{\hat{z}}\hat{g}$, with the relations on their indices as follows

$$(D_z g)_{i,j} \neq 0 \implies (D_{\hat{z}}\hat{g})_{i,\pi(j)} \neq 0. \tag{61}$$

Furthermore, because of the assumption that

$$\|D_{\hat{Z}}\hat{g}\|_0 \leq \|D_Z g\|_0, \tag{62}$$

Eq. (61) can be further restricted to an equivalence between the sparsity patterns, i.e.,

$$(D_z g)_{i,j} \neq 0 \Longleftrightarrow (D_{\hat{z}}\hat{g})_{i,\pi(j)} \neq 0. \tag{63}$$

Therefore, there must be

$$\text{supp}(D_z g) = \text{supp}((D_{\hat{z}}\hat{g})P), \tag{64}$$

where $P$ denotes a permutation matrix. Thus, the support of the Jacobian matrix $D_{\hat{z}}\hat{g}$ is identical to that of $D_z g$, up to a permutation of column indices. $\qquad \square$

## A.4   PROOF OF THEOREM 3

**Theorem 3** (**Element identifiability**). *Consider any two models $\theta = (g, p_Z)$ and $\hat{\theta} = (\hat{g}, p_{\hat{Z}})$ following the process in Sec. 2. Suppose assumptions in Thm. 1 and Assum. 2 hold. Then we have identifiability up to element-wise indeterminacy, i.e., $\theta \sim_{obs} \hat{\theta} \implies \theta \sim_{elem} \hat{\theta}$.*

*Proof.* Since all assumptions in Thm. 1 are satisfied, for these two models $\theta = (g, p_Z)$ and $\hat{\theta} = (\hat{g}, p_{\hat{Z}})$ following the process in Sec. 2, we can follow the same steps in Sec. A.2 to derive Eq. (21), i.e.,

$$(D_z g)_{i,j} \neq 0 \iff (D_{\hat{z}} \hat{g})_{i,\pi(j)} \neq 0. \tag{65}$$

Then, for any latent varible $Z_i \in Z$, let us consider all conditions in Assum. 2. We begin with the first condition: there exists a set of observed variables $A$ and an element $X_k \in A$ such that

$$\bigcup_{X_j \in A} I_j = [d_z], \quad \text{and} \quad I_k \setminus \bigcup_{X_j \in A \setminus \{X_k\}} I_j = \{i\}. \tag{66}$$

Our want to show that, for any other $r \neq i$, we have

$$\frac{\partial Z_i}{\partial \hat{Z}_{\pi(r)}} = 0. \tag{67}$$

We consider two cases:

- $r \in \left( \bigcup_{X_j \in A \setminus \{X_k\}} I_j \right) \setminus I_k$;

- $r \in \left( \bigcup_{X_j \in A \setminus \{X_k\}} I_j \right) \cap I_k$.

Suppose $r \in \left( \bigcup_{X_j \in A \setminus \{X_k\}} I_j \right) \setminus I_k$. Let us denote $J_{A \setminus k}$ as the index set of $A \setminus \{X_k\}$. Since $I_k \setminus \bigcup_{X_j \in A \setminus \{X_k\}} I_j = \{i\}$, for any $v \in J_{A \setminus k}$, there must be

$$i \notin \mathrm{supp}(D_z g)_{v,.}, \tag{68}$$

We then suppose for contradiction that

$$M_{\phi_{i,.}} \in \mathrm{span}\{e_l : l \in \mathrm{supp}((D_{\hat{Z}} \hat{g})_{v,.})\}. \tag{69}$$

In the proof of Theorem 1, we have proved that

$$M_{\phi_{i,\pi(i)}} \neq 0. \tag{70}$$

Then we have

$$\pi(i) \in \mathrm{supp}(D_{\hat{Z}} \hat{g})_{v,.} \tag{71}$$

According to Eq. (65), this implies

$$i \in \mathrm{supp}(D_Z g)_{v,.} \tag{72}$$

This contradicts

$$i \notin \mathrm{supp}(D_Z g)_{v,.} \tag{73}$$

Thus, there must be

$$M_{\phi_{i,.}} \notin \mathrm{span}\{e_l : l \in \mathrm{supp}((D_{\hat{Z}} \hat{g})_{v,.})\} \tag{74}$$

We further suppose by contradiction that

$$M_{\phi_{i,\pi(r)}} \neq 0, \tag{75}$$

for $r \in \left( \bigcup_{X_j \in A \setminus \{X_k\}} I_j \right) \setminus I_k$. Then, according to Eq. (74), there must be

$$\pi(r) \notin \mathrm{supp}(D_{\hat{Z}} \hat{g})_{v,.}, \tag{76}$$

which implies

$$r \notin \mathrm{supp}(D_Z g)_{v,.} \tag{77}$$

This is, again, a contradiction to $r \in \left( \bigcup_{X_j \in A \setminus \{X_k\}} I_j \right) \setminus I_k$. As a result, there must be

$$M_{\phi_{i,\pi(r)}} = 0. \tag{78}$$

We then consider the other case, where we assume $r \in (\bigcup_{X_j \in A \setminus \{X_k\}} I_j) \cap I_k$. Then there exists $q \in J_{A \setminus k}$ s.t.

$$i \in \text{supp}(D_Z g)_{q,\cdot}, \tag{79}$$

which further implies

$$M_{\phi_{i,\cdot}} \in \text{span}\{e_{q'} : q' \in \text{supp}((D_{\hat{Z}} \hat{g})_{q,\cdot})\}. \tag{80}$$

Since we also have

$$i \notin \text{supp}(D_Z g)_{q,\cdot}. \tag{81}$$

We suppose for contradiction that

$$M_{\phi_{i,\cdot}} \in \text{span}\{e_l : l \in \text{supp}((D_{\hat{Z}} \hat{g})_{q,\cdot})\}. \tag{82}$$

Since there is

$$M_{\phi_{i,\pi(i)}} \neq 0. \tag{83}$$

It follows that

$$\pi(i) \in \text{supp}(D_{\hat{Z}} \hat{g})_{q,\cdot}. \tag{84}$$

According to Eq. (65), it implies

$$i \in \text{supp}(D_Z g)_{q,\cdot}. \tag{85}$$

This contradicts the case that $i \notin \text{supp}(D_Z g)_{q,\cdot}$, and thus there must be

$$M_{\phi_{i,\cdot}} \notin \text{span}\{e_l : l \in \text{supp}((D_{\hat{Z}} \hat{g})_{q,\cdot})\}. \tag{86}$$

For $r \in (\bigcup_{X_j \in A \setminus \{X_k\}} I_j) \cap I_k$, suppose

$$M_{\phi_{i,\pi(r)}} \neq 0. \tag{87}$$

Given Eqs. (80) and (86), we have

$$\pi(r) \in \text{supp}(D_{\hat{Z}} \hat{g})_{i,\cdot}, \tag{88}$$

$$\pi(r) \notin \text{supp}(D_{\hat{Z}} \hat{g})_{q,\cdot}. \tag{89}$$

Because of Eq. (65), these further imply

$$r \in \text{supp}(D_Z g)_{i,\cdot}, \tag{90}$$

$$r \notin \text{supp}(D_Z g)_{q,\cdot}. \tag{91}$$

This leads to

$$r \in I_k \setminus \bigcup_{X_j \in A \setminus \{X_k\}} I_j, \tag{92}$$

which contradicts

$$r \in \left( \bigcup_{X_j \in A \setminus \{X_k\}} I_j \right) \cap I_k. \tag{93}$$

Therefore, there must be

$$M_{\phi_{i,\pi(r)}} = 0. \tag{94}$$

Since $M_\phi$ is the support of $D_{\hat{Z}} \phi^{-1}$, this implies that, for any other $r \neq i$, we have

$$\frac{\partial Z_i}{\partial \hat{Z}_{\pi(r)}} = 0. \tag{95}$$

Because $\phi$ is invertible, each row of $D_{\hat{Z}} \phi^{-1}$ must at least have one non-zero element. Therefore, it follows that

$$\frac{\partial Z_i}{\partial \hat{Z}_{\pi(i)}} = 0. \tag{96}$$

Thus, with the first condition in Assum. 2, we have identifiability up to element-wise indeterminacy.

Next, we consider the second condition in Assum. 2. By applying the case of *Intersection* in Defn. 5 for all pairs of observed variables in $X$, there is

$$\frac{\partial Z_{\bigcap_{X_j \in A \setminus \{X_k\}} I_j}}{\partial \sigma(\hat{Z})_{\Delta_{X_j \in A \setminus \{x_k\}} I_j}} = 0, \tag{97}$$

where $\sigma$ denotes the transformation for the permutation $pi$. Then for $\bigcup_{X_j \in A \setminus \{X_k\}} I_j$ and $I_k$, by the *individual-centric disentanglement* in Prop. 1, there is

$$\frac{\partial Z_{\left(\bigcup_{X_j \in A \setminus \{X_k\}} I_j\right) \setminus I_k}}{\partial \sigma(\hat{Z})_{I_k}} = 0. \tag{98}$$

Note that

$$\left(Z_{\bigcap_{X_j \in A \setminus \{X_k\}} I_j}\right) \cap \left(Z_{\left(\bigcup_{X_j \in A \setminus \{X_k\}} I_j\right) \setminus \{X_k\}}\right) = Z_{\left(\bigcap_{X_j \in A \setminus \{X_k\}} I_j\right) \setminus I_k} \tag{99}$$

Considering both Eqs. (97) and (99), we have

$$\frac{\partial Z_{\left(\bigcap_{X_j \in A \setminus \{X_k\}} I_j\right) \setminus I_k}}{\partial \sigma(\hat{Z})_{\Delta_{X_j \in A \setminus \{X_k\}} I_j}} = 0. \tag{100}$$

Considering both Eqs. (98) and (99), we have

$$\frac{\partial Z_{\left(\bigcap_{X_j \in A \setminus \{X_k\}} I_j\right) \setminus I_k}}{\partial \sigma(\hat{Z})_{I_k}} = 0. \tag{101}$$

Note that

$$\sigma(\hat{Z})_{\Delta_{X_j \in A \setminus \{X_k\}} I_j} \cup \sigma(\hat{Z})_{I_k} \tag{102}$$

$$= [d_z] \setminus \left(\left(\bigcap_{X_j \in A \setminus \{X_k\}} I_j\right) \setminus I_k\right) \tag{103}$$

$$= [d_z] \setminus i. \tag{104}$$

Further given the invertibility of $\phi$, each row of $D_{\hat{Z}}\phi^{-1}$ must at least have one non-zero element. Therefore, it follows that

$$\frac{\partial Z_i}{\partial \hat{Z}_{\pi(i)}} \neq 0. \tag{105}$$

Lastly, we consider the third condition in Assum. 2. That part of proof directly follows from (Lachapelle et al., 2022; Zheng et al., 2022). Suppose for each row in $M_\phi$, there are more than one non-zero element. Then

$$\exists j_1 \neq j_2, M_{\phi_{j_1,.}} \cap M_{\phi_{j_2,.}} \neq \emptyset. \tag{106}$$

Then consider $j_3 \in [d_z]$ such that

$$\pi(j_3) \in M_{\phi_{j_1,.}} \cap M_{\phi_{j_2,.}}. \tag{107}$$

Since $j_1 \neq j_3$, it is either $j_3 \neq j_1$ or $j_3 \neq j_2$. Without loss of generality, we assume $j_3 \neq j_1$.

Since we have

$$\bigcap_{X_j \in A_{j_1}} I_j = j_1, \tag{108}$$

there must exists $X_{i_3} \in A_{j_1}$ such that $j_3 \neq I_{i_3}$. Because $j_1 \in I_{i_3}$, we have

$$(i_3, j_1) \in \text{supp}(D_Z g), \tag{109}$$

which further implies

$$M_{\phi_{j_1,.}} \in \text{span}\{e'_k : k' \in \text{supp}((D_{\hat{Z}}\hat{g})_{i_3,.})\}. \tag{110}$$

Given Eq. ([107](#)), it implies

$$\pi(j_3) \in \text{supp}(D_{\hat{Z}}\hat{g})_{i_3,\cdot}. \tag{111}$$

This, again, implies

$$j_3 \in \text{supp}(D_Z g)_{i_3,\cdot}, \tag{112}$$

which contradicts $j_3 \neq I_{i_3}$. Therefore, for each row in $M_\phi$, there are no more than one non-zero element. Because $M_\phi$ is invertible. each row must at least have one non-zero element. Thus, there must be exactly one non-zero element each row, which is

$$\frac{\partial Z_i}{\partial \hat{Z}_{\pi(i)}} \neq 0. \tag{113}$$

Thus, we have proved our goal with all three conditions. $\qquad\square$

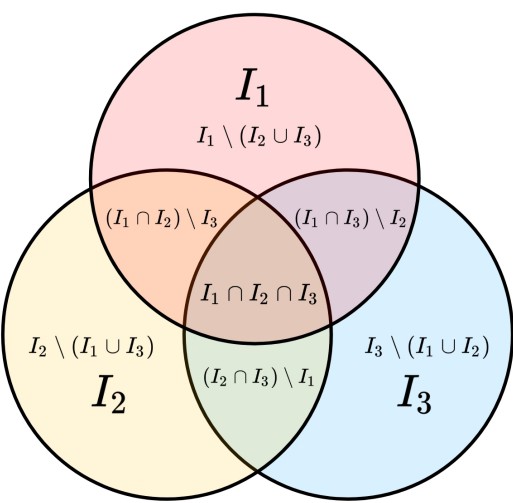

Figure 6: The Venn diagram example (Fig. 3).

## B    ADDITIONAL DISCUSSION

### B.1    THE VENN DIAGRAM.

Here we provide the full deriviation of the Venn diagram example.

**Example 5** (Identifying all atomic regions). *Let $I_1$, $I_2$, and $I_3$ be the latent index sets of $X_1$, $X_2$, and $X_3$ in Fig. 3. For each atomic region $\mathcal{A}$ we pick two sets of observed variables $(X_K, X_V)$ so that every $i \in \mathcal{A}$ satisfies one of the three conditions in Defn. 5 with every $j \notin \mathcal{A}$. This guarantees that the latents in $\mathcal{A}$ are disentangled from all others, establishing block-wise identifiability.*

**(i)** $I_1 \setminus (I_2 \cup I_3)$    *Step 1: Take $X_K = X_1$, $X_V = X_2$. Then $i \in I_K \setminus I_V$ and every $j \in I_2$ lies in $I_V \setminus I_K$, so case (iii) applies. Step 2: Take $X_K = X_1$, $X_V = X_3$. Now $i \in I_K \setminus I_V$ and every $j \in I_3$ is in $I_V \setminus I_K$, again case (iii). All indices outside $\mathcal{A}$ belong to $I_2$ or $I_3$ (or both), so $\mathcal{A}$ is disentangled.*

**(ii)** $I_2 \setminus (I_1 \cup I_3)$    *Symmetric to (i) with the roles of $(1,2)$ and $(2,1)$ swapped: use $(X_K, X_V) = (X_2, X_1)$ and $(X_2, X_3)$.*

**(iii)** $I_3 \setminus (I_1 \cup I_2)$    *Symmetric to (i): use $(X_K, X_V) = (X_3, X_1)$ and $(X_3, X_2)$.*

**(iv)** $(I_1 \cap I_2) \setminus I_3$    *This is the worked example already given; we recap for completeness. Step 1: $(X_1, X_2)$ yields $i \in I_K \cap I_V$, $j \in I_K \Delta I_V$ (case (ii)). Step 2: $(X_1 \cup X_2, X_3)$ yields $i \in I_K \setminus I_V$, $j \in I_V$ (case (iii)).*

**(v)** $(I_1 \cap I_3) \setminus I_2$    *Step 1: $X_K = X_1$, $X_V = X_3$: $i \in I_K \cap I_V$, $j \in I_K \Delta I_V$ (case (ii)). Step 2: $X_K = X_1 \cup X_3$, $X_V = X_2$: $i \in I_K \setminus I_V$, $j \in I_V$ (case (iii)).*

**(vi)** $(I_2 \cap I_3) \setminus I_1$    *Step 1: $X_K = X_2$, $X_V = X_3$: $i \in I_K \cap I_V$, $j \in I_K \Delta I_V$ (case (ii)). Step 2: $X_K = X_2 \cup X_3$, $X_V = X_1$: $i \in I_K \setminus I_V$, $j \in I_V$ (case (iii)).*

**(vii)** $I_1 \cap I_2 \cap I_3$    *Step 1: $X_K = X_1$, $X_V = X_2$: $i \in I_K \cap I_V$, any $j$ that differs only by presence in $I_1$ or $I_2$ lies in $I_K \Delta I_V$ (case (ii)). Step 2: $X_K = X_1 \cup X_2$, $X_V = X_3$: $i \in I_K \cap I_V$, while every remaining $j$ either (a) appears in exactly one of $I_1, I_2, I_3$ and so is in $I_K \Delta I_V$ (case (ii)), or (b) lies solely in $I_3$ and is in $I_V \setminus I_K$ (case (iii)).*

*In every case the chosen pairs cover all $j \notin \mathcal{A}$, so each atomic region is disentangled from the rest and hence block-wise identifiable under the invertibility assumption.*

B.2 FURTHER DISCUSSION ON THE CONNECTION OF CONDITIONS.

The spirit of many of our conditions stems from the classical literature of latent variable models. Here we discuss the connections in more details:

**Connection between diversity and sparsity assumptions.** The structural diversity condition is firstly introduced in our work for the nonlinear case, but other conditions on the structure appear in related field, e.g., the sparsity condition. For instance, the anchor feature assumption (Arora et al., 2012; Moran et al., 2021), structural sparsity Zheng et al. (2022), sparse dictionary (Garfinkle & Hillar, 2019), and many others. Although both diversity and sparsity concern latent structure, they are fundamentally different. Diversity focuses on variation in the dependency between latent and observed variables, and does not require sparsity at all. It remains valid even in nearly fully connected graphs, as long as there is some variation (e.g., even a single differing edge) in the connectivity patterns across variables. In contrast, sparsity conditions do not imply diversity and always enforce sparse connectivities.

**Connection between injectivity and RIP assumptions.** Injectivity is a standard requirement to ensure that latent information is not lost through the generative map. In the linear setting, this reduces to classical Restricted Isometry Property (RIP) conditions (Foucart & Rauhut, 2013; Jung et al., 2016), which are known to be necessary for linear dictionary learning. Our injectivity condition is the natural nonlinear analogue: it prevents degenerate mappings that collapse latent variation, standard in almost all previous work on nonparametric identifiability (Hyvärinen et al., 2024; Moran & Aragam, 2025).

**Connection between dependency sparsity and latent sparsity regularizations.** Beyond assumptions on the data generating process, there are also connections in terms of regularization during estimation. Most prior work on sparse dictionary learning imposes sparsity directly on the recovered latent variables $Z$. The most prominent example is the Sparse Autoencoder (SAE), which is based on sparse dictionary learning and widely used in mechanistic interpretability. However, as highlighted by recent reviews (Cunningham et al., 2023), latent sparsity causes issues such as feature absorption and extremely high latent dimensionality (e.g., millions of variables). In contrast, we regularize sparsity on the dependency structure (Jacobian sparsity) rather than the latents themselves, which avoids these issues and has shown benefits both in our own experiments (Sec. 4.2) and in recent work on large language models, such as the Jacobian Sparse Autoencoder (Farnik et al., 2025).

B.3 FURTHER DISCUSSION ON THE CONNECTION WITH SAES.

**Discussion.** Mechanistic interpretability often seeks to uncover the underlying concepts that drive the behavior of large language models, whether to understand sources of hallucination or to explain model responses. Sparse Autoencoders (SAEs) have been widely used for this purpose, but, as noted in the community [3], SAEs rely on sparse dictionary learning and therefore inherit its limitations:

- First, SAEs fundamentally assume a linear generative function, since sparse dictionary learning lacks nonlinear identifiability. As you mentioned, the Linear Representation Hypothesis (LRH) is a reasonable working assumption in many contexts, but LRH concerns the linear relation within the latent space of $Z$ rather than the linearity of the generative map $g$ in $X = g(Z)$. In reality, most models have highly nonlinear generative functions (for example, due to nonlinear activations such as ReLU or GeLU). Assuming linearity at the generative level simplifies implementation but inevitably introduces bias and prevents full recovery of the true latent factors in these nonlinear settings.

  Different from the theoretical foundation of SAEs (i.e., sparse dictionary learning), our diverse dictionary learning provides theoretical guarantees for nonlinear latent variable models, offering a provably rigorous tool for mechanistic interpretability in almost all real-world scenarios, covering the previously unsupported nonlinear cases.

- Second, SAEs impose sparsity directly on the latent variables. This often forces extremely high-dimensional latent spaces (e.g., millions of units) to represent real-world concepts. More, latent sparsity can cause feature splitting and absorption, where important concepts are lost due to the encouragement of sparse latents. For instance, if we only maximize the

latent sparsity, each feature will only corresponds to several samples, capture concepts that are very specific (e.g., persian cats) versus more general concepts (e.g., cats).

In contrast, diverse dictionary learning encourages dependency sparsity (Jacobian sparsity) rather than latent sparsity, and is designed for nonlinear generative models with identifiability guarantees. This provides a principled solution to both limitations of SAEs and has meaningful implications for mechanistic interpretability.

**Empirical Evidence.** The recent Jacobian Sparse Autoencoder (JSAE) work (Farnik et al., 2025) provides extensive empirical support for the same dependency sparsity regularization required by our diverse dictionary learning theory. Their results suggest that replacing traditional SAE losses with dependency sparsity–based losses can improve both interpretability and efficiency. We therefore refer readers in mechanistic interpretability to their thorough empirical study.

At the same time, there is one perspective that (Farnik et al., 2025) has not evaluated against other SAE baselines, i.e., the number of dead features. To make the empirical evedeinces even more comprehensive, we conducted new experiments on the OpenWebtext with GPT2-Small, comparing JSAE with Top-K SAE (Gao et al., 2025) and Batch Top-K SAE (Bussmann et al.), with the latent dimension as $12,288$. The results are in Table 3.

Table 3: Number of dead features under different methods

| Method | # of Dead Features |
|---|---|
| Top-K SAE | 439 |
| Batch Top-K | 207 |
| JSAE | 62 |

These results, together with the comprehensive experiments in (Farnik et al., 2025), further support the advantages of dependency sparsity, demonstrating that it not only yields more interpretable representations but also preserves active and meaningful latent features more effectively.

### B.4 FURTHER DISCUSSION ON SUFFICIENT NONLIENARITY

The sufficient nonlinearity assumption is meant to ensure that the Jacobian varies enough across samples so that its Jacobian vectors span the relevant support. Although this condition may seem demanding at first glance, it is usually quite mild in practice. For each observed variable $X_i$, the requirement involves only $|(D_Z g)_{i,\cdot}|_0$ samples, which is simply the number of latent variables that influence $X_i$, and this number is typically far smaller than the available sample size.

When $g$ is smooth and the latent distribution has a continuous density, Jacobian evaluations at independently drawn samples form continuous random vectors. Such vectors are in general position with probability one, which means that a small number of random samples already produces Jacobian rows that span the required support. For example, if $X_i$ depends on five latent coordinates, then roughly five random samples are usually sufficient, even in much higher dimensional systems.

The second part $\operatorname{supp}((D_Z g(z^{(k)})H)_{i,\cdot}) \subseteq \operatorname{supp}((D_{\hat{Z}}\hat{g})_{i,\cdot})$ avoids the cases where true dependencies being diminished across *all* samples during estimation. We have $(D_{\hat{Z}}\hat{g}(\hat{z}))_{i,\cdot} = (D_Z g(z))_{i,\cdot} h(z,\hat{z})$, which already lies inside $\operatorname{supp}((D_{\hat{Z}}\hat{g})_{i,\cdot})$. The assumption requires the existence of several samples in the whole space where that relation holds for a matrix of the same support as $h$. The key nuance is that the condition only require that matrix exists for several samples of $z$ in the whole continuous space, but identifiability is defined as an asymptotic property over infinite samples. Conceptually, this condition implies that the estimation should have a denser Jacobian compared to the ground truth. It rules out the case where a true dependency is globally eliminated during estimation. In practice, we start with a neural network, which typically has a fully dense Jacobian when initialized, and then applying sparsity regularization to shave away redundant dependencies. A violation of this assumption would therefore correspond to a global cancellation of a genuine dependency across all samples, which is intuitively similar to a violation of faithfulness. As a result, alternative conditions may also be applied to achieve similar goals.

### B.5 DEPENDENCY SPARSITY REGULARIZATION ON LARGE MODELS

Computing the full Jacobian of a large model can be expensive. Fortunately, recent work (Farnik et al., 2025) shows that dependency sparsity regularization remains practical even at scale up to common LLMs when combined with two standard strategies.

- **Apply latent sparsity first.** Large models often have high dimensional latent spaces, but for any given input, many latent coordinates are inactive or irrelevant. A common approach is to first identify the active coordinates and compute the Jacobian only with respect to this subset. Since the active block is often tiny compared to the full latent space, this reduces both computation and memory by several orders of magnitude in typical transformer architectures [1]. In practice, the Jacobian is rarely formed as a dense $d_x \times d_z$ matrix, but only as a restricted slice that corresponds to the active latent directions for that specific input.

- **Use efficient closed-form expressions.** For several widely used architectures, the Jacobian with respect to selected latent directions has efficient factorizations. As shown in [1], models with residual attention and feedforward structure admit closed form expressions for the relevant Jacobian blocks that require only a few matrix multiplications and inexpensive elementwise operations. This avoids repeated backward passes through the full model and keeps the cost manageable even when the underlying network is large.

With these strategies, training a large model with our dependency sparsity regularization is reported to be only about twice as slow as training with standard $\ell_1$ regularization on the latent variables $Z$ (Farnik et al., 2025).

### B.6 GENERAL NOISE AND NON-INVERTIBILITY

In our paper, we follow the standard setup in the identifiability literature regarding noise and invertibility because violations of it usually require much stronger assumptions to compensate. That said, we feel that more discussion on how to handle these cases is helpful in certain scenarios.

**Handling general noises.** Most identifiability results for latent variable models focus on additive independent noise or noiseless settings. This holds for both classical linear models (Reiersøl, 1950) and recent nonparametric work (Hyvärinen et al., 2024). The main difficulty is separating noise from latent variables, since in the general form they can be entangled in complex ways. Recent work (Zheng et al., 2025), based on the Hu-Schennach theorem (Hu & Schennach, 2008), shows that general noise can be separated under additional assumptions on the generative function $f$ and on conditional independence across groups of observed variables. As noted in Remark 1, under the same conditions, our results extend to settings with general noise.

**Handling partial non-invertibility.** Invertibility and its variants are among the most common assumptions in the literature. Without additional assumptions, it can even be necessary, since information that is lost during generation cannot be recovered in principle. In scenarios where partial non-invertibility must be addressed, one may consider incorporating temporal information together with sufficient changes in the nonstationary transition (Chen et al., 2024), which can provide the extra information needed for recovery. Since our theory focuses on the general setting without any auxiliary information, we adopt the standard assumption of invertibility.

### B.7 FURTHER DISCUSSION ON POTENTIAL IMPACT.

In a nutshell, recovering the ground-truth data generative process is important for both predictive and non-predictive tasks. Machine learning is fundamentally a balance between inductive bias and data. By uncovering the hidden world underlying the data, we obtain principled, domain-agnostic inductive biases grounded in fundamental understanding rather than heuristics. These insights can be embedded directly into model architectures, training objectives, and evaluation protocols, yielding systems that are more robust, generalizable, and data-efficient. Importantly, these inductive biases are not merely theoretical; they have already produced measurable improvements across diverse real-world domains.

We first highlight two overarching benefits:

**Truthfulness.** It is well known that, given any pair of observationally equivalent models (e.g., models perfectly trained by MLE), without additional assumptions, there is no guarantee at all that these models actually recover the underlying data generative processes. This has led to the critical issue of non-identifiability of the nonlinear latent variable model (e.g., the classical work on the

non-uniqueness of nonlinear ICA (Hyvärinen & Pajunen, 1999)), and has been widely validated by large-scale empirical studies on unsupervised learning (e.g., (Locatello et al., 2019)). This raises significant challenges for tasks (e.g., transfer learning, controllable generation, compositional generalization, and mechanistic interpretability) where we need to understand the true process, rather than purely fitting the observed distribution. Therefore, understanding what aspects can still be recovered, and what inductive biases should be introduced to guide recovery, provide both theoretical guarantees and practical guidance on ensuring the truthfulness of the learning. These are exactly what our paper aims to address.

**Efficiency.** Recovering the underlying true generative factors provides a principled solution to improve the efficiency, since we only need to model those essential representation capturing what we are interested, instead of the whole high-dimensional space that also include numerous irrelevant information or arbitrary noises. For instance, if we can make sure that the recovered latents will not be entangled with other latent variables, we can precisely only on those that relevant to our tasks (e.g., content for transfer learning, or specific latent concepts we need to modify) instead of the whole latent spaces, not only reducing the cost but enforcing the model to focus only on those relevant, improve the efficiency in a principled way.

We have conducted experiments on visual disentanglement to illustrate one of the potential application scenarios. Across datasets and backbone models, the results consistently show that adding the identifiability-guided regularization (dependency sparsity) enables the model to recover the underlying generative factors. This not only improves disentanglement performance, which is itself practically important, but also directly benefits the following application scenarios.

**Mechanistic Interpretability.** In many scenarios, we aim to study the underlying concepts that generate the responses of LLMs, for many reasons such as investigating the root of hallucination or explaining the mechanisms for specific responses or behaviors. This is related to the field of Mechanistic Interpretability, where Sparse Autoencoder (SAE) has been popular. However, as known by the community (Sharkey et al., 2025), SAE is based on the theory of Sparse Dictionary Learning, and there are some open problems:

First, SAE can only deal with linear data due to the lack of nonlinear identifibality of sparse dictioanry learning. However, the real-world is full of nonlinearity. Assuming everything is linear simplicity the procedure and will unavoidably bring bias and cannot fully recover the truth.

Moreover, SAE encourages sparsity over the latent variables. As a result, it needs an extremely high-dimensional vector to capture real-world concepts (e.g., millions of dimensions), and some features are easy to be absorbed due to its encourage on latent sparsity. In contrast, our Diverse Dictionary Learning encourage the sparsity on the Jacobian (i.e., dependency sparsity) instead of latent diversity, and can handle nonlinear generative processes with identiability guarantees. This provides a principled solution to both open problems of SAE, which is highly significant for the whole filed of mechanistic interpretability.

**Transfer Learning.** In transfer learning, it is important to disentangle the invariant and changing part, such as disentangling invariant content from changing styles. Identifiability has been widely leveraged in the previous literature on guaranteeing reliable and efficient domain adaptation (e.g., (Von Kügelgen et al., 2021; Kong et al., 2022; Li et al., 2023)). Given two observed variables, our generalized identifiability results guarantees the recovery of the shared and private parts of their latent variables, under a simple regularization of dependency sparsity. This provides a flexible framework for transfer learning that can really capture the essential part across domains.

**Controllable Generation.** A perfect prediction machine can only master at mining correlations, while leveraging the true causation relies on identifiability. For instance, if we want to add eyeglasses on a kid's face, due to the overwhelming correlation between wearing eyeglasses and the relatively higher age, models sometimes will also increase the age of the kid, even for large foundational models. This negligence of the true underlying process is one of the fundamental reasons on why large models, although extensively trained on web-scale data, can still provide many extra changes that are out of our control. Previous work has already shown that, by incorporating inductive bias guided via identifiability, models can achieve precise control on the generated images (Xie et al., 2025a; 2023).

| Dim | Ours | OroJAR | Hessian Penalty |
|---|---|---|---|
| 3 | $0.8258 \pm 0.0085$ | $0.7288 \pm 0.0280$ | $0.8257 \pm 0.0240$ |
| 4 | $0.8449 \pm 0.0043$ | $0.6301 \pm 0.0810$ | $0.8352 \pm 0.0396$ |
| 5 | $0.8048 \pm 0.0080$ | $0.5119 \pm 0.1482$ | $0.7789 \pm 0.0174$ |

Table 4: MCC under different regularization penalties across dimensions (mean $\pm$ std, higher is better).

| Dim | Ours w/o noise | Ours w/ noise | Base |
|---|---|---|---|
| 3 | $0.8258 \pm 0.0085$ | $0.8210 \pm 0.0088$ | $0.3814 \pm 0.0369$ |
| 4 | $0.8449 \pm 0.0043$ | $0.8381 \pm 0.0093$ | $0.5467 \pm 0.0326$ |
| 5 | $0.8048 \pm 0.0080$ | $0.7944 \pm 0.0134$ | $0.4576 \pm 0.1075$ |

Table 5: MCC across dimensions with and without noise (mean±std, higher is better).

**Multi-modal Alignment.** Similarly, one may consider multiple modalities as multiple sets of observed variables, where the semantically meaningful concepts are those that shared by multiple modalities, and the modality-specific concepts (e.g., texture for images, volumns for audio) are those private latent variables. By formalizing the process as a general dictionary learning problem, many previous work have already shown the practical implication of identifiability, such as addressing information misalignment (Xie et al., 2025b) and capturing complex interactions across different modalities (Sun et al., 2024).

**Scientific Discovery.** Understanding the hidden truth from observation is always one of the main tasks of scientific discovery. Newton observed the falling apple (observation $X$), then he got curious, studied much, and found out it was actually gravity (latent variables $Z$) causing items to fall. Therefore, the fundamental task of scientific discovery may be sumamrized into the simple equation of $X = f(Z)$, where we aim to recover $Z$ based solely on $X$, which is exactly our task of general dictionary learning. Of course, the guarantee that the recovered latent variables $\hat{Z}$ correspond to the ground-truth latents $Z$ in a meaningful way is what matters the most, and this is exactly the focus of our generalized identifiability theory. There have been numerous successful applications of identifiable representation leraning in the literature, such as dynamic systems like climate change (Yao et al., 2024a), robotics (Lippe et al., 2023), neuralimaging (Hyvärinen & Morioka, 2016) and genomics (Morioka & Hyvarinen, 2024).

Of course, the list is non-exclusive, and will only grow faster given the development of large-scale models. The reason is simple: even with infinite data and computation, without identifiability, models may achieve perfect predictions but cannot be guaranteed to recover the underlying truth. Thus, the closer we are to that boundary, the more efforts we should put into going beyond correlation.

## C  ADDITIONAL EXPERIMENTS

In this section, we present further experiments on both synthetic and real-world data.

### C.1  ADDITIONAL SYNTHETIC EXPERIMENTS

We begin with a series of additional experiments in the synthetic setting.

**Additional baselines.** We first compare dependency sparsity to alternative regularizers. Table 4 reports MCC for $d \in \{3, 4, 5\}$ against two Jacobian/Hessian penalties: OroJAR (Wei et al., 2021) and the Hessian Penalty (Peebles et al., 2020). Neither provides identifiability guarantees in the non-parametric setting. Empirically, both underperform our method, with a widening gap as $d$ increases. This indicates that penalizing the dependency map in a structural way improves recovery of the true latent factors.

**Noise robustness.** Remark 1 states that our framework naturally extends to generative processes with additive noise. Table 5 confirms this: MCC remains essentially unchanged compared to the

| $\lambda$ | Dimensionality | | |
|---|---|---|---|
| | 3 | 4 | 5 |
| 0 | $0.6789 \pm 0.0364$ | $0.7317 \pm 0.1092$ | $0.6989 \pm 0.0133$ |
| 0.001 | $0.7313 \pm 0.0242$ | $0.7294 \pm 0.0147$ | $0.7513 \pm 0.0007$ |
| 0.005 | $0.7765 \pm 0.0363$ | $0.7826 \pm 0.0244$ | $0.7681 \pm 0.0455$ |
| 0.01 | $0.8145 \pm 0.0221$ | $0.8032 \pm 0.0327$ | $0.7979 \pm 0.0421$ |
| 0.03 | $0.8268 \pm 0.0268$ | $0.8232 \pm 0.0187$ | $0.8101 \pm 0.0112$ |
| 0.05 | $0.8256 \pm 0.0088$ | $0.8420 \pm 0.0401$ | $0.8099 \pm 0.0296$ |

Table 6: MCC across different $\lambda$ values (sparsity regularization weight) and dimensionalities (mean±std, higher is better).

| Method | FactorVAE ↑ | DCI ↑ |
|---|---|---|
| FactorVAE | $0.708 \pm 0.026$ | $0.135 \pm 0.030$ |
| + Latent Sparsity | $0.501 \pm 0.434$ | $0.113 \pm 0.069$ |
| + Dependency Sparsity | $\mathbf{0.752 \pm 0.040}$ | $\mathbf{0.144 \pm 0.053}$ |
| + Dependency Sparsity (128) | $0.723 \pm 0.023$ | $0.141 \pm 0.004$ |

Table 7: Quantitative comparison on Cars3D dataset (mean±std, higher is better).

noiseless case, with only minor drops, while the base model degrades sharply. This supports the claim that dependency sparsity stabilizes latent recovery under noise. Extending identifiability to arbitrary noise remains more challenging in the nonparametric setting, since invertibility can break down and stronger assumptions are typically required.

**Regularization weight.** To examine the effect of regularization strength, we vary the sparsity weight $\lambda$ in Table 6. MCC increases steadily from $\lambda = 0$ and plateaus around $\lambda \in [0.03, 0.05]$, showing that the method is stable and not overly sensitive once past the under-regularized regime. Importantly, sparsity here serves only as an inductive bias during estimation. Our theory does not assume the data-generating process itself is sparse. Instead, it relies on structural diversity, which can hold even in dense settings. Moreover, the set-theoretic framework is robust to partial violations of assumptions and still enables meaningful recovery when full identifiability is unattainable.

### C.2 ADDITIONAL VISUAL EXPERIMENTS

We next evaluate more on images, providing both quantitative comparisons and qualitative analyses.

**Scalability.** To assess scalability, we upsampled Cars3D to $128 \times 128$ and re-ran FactorVAE with dependency sparsity. As shown in Table 7 (last row), performance remains consistent with the $64 \times 64$ setting. This suggests that the observed improvements stem from leveraging structural regularization rather than resolution, and that the method scales robustly with image size.

**Quantitative evaluation.** Table 7 evaluates FactorVAE score and DCI on Cars3D. Adding *dependency* sparsity improves FactorVAE from $0.708$ to $0.752$ and DCI from $0.135$ to $0.144$. Latent sparsity often underperforms. Table 8 extends to MPI3D and includes OroJAR and the Hessian

| Method | Cars3D | | MPI3D | |
|---|---|---|---|---|
| | FactorVAE ↑ | DCI ↑ | FactorVAE ↑ | DCI ↑ |
| FactorVAE | $0.708 \pm 0.026$ | $0.135 \pm 0.030$ | $0.599 \pm 0.064$ | $0.345 \pm 0.047$ |
| + Latent Sparsity | $0.501 \pm 0.434$ | $0.113 \pm 0.069$ | $0.440 \pm 0.065$ | $0.325 \pm 0.028$ |
| + OroJAR | $0.165 \pm 0.235$ | $0.030 \pm 0.007$ | $0.499 \pm 0.090$ | $0.272 \pm 0.054$ |
| + Hessian Penalty | $0.321 \pm 0.455$ | $0.082 \pm 0.077$ | $0.506 \pm 0.056$ | $0.254 \pm 0.067$ |
| **+ Dependency Sparsity** | $\mathbf{0.752 \pm 0.040}$ | $\mathbf{0.144 \pm 0.053}$ | $\mathbf{0.639 \pm 0.084}$ | $\mathbf{0.384 \pm 0.031}$ |

Table 8: Quantitative comparison on Cars3D and MPI3D datasets (mean±std, higher is better).

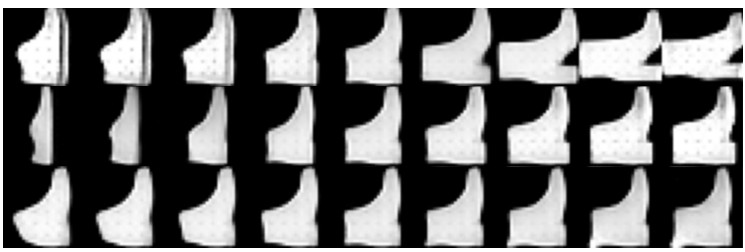

Figure 7: Latent variable visualization on Fashion with Flow + Dependency Sparsity. From top to bottom, the latent variables correspond to gender, heel height, and upper width.

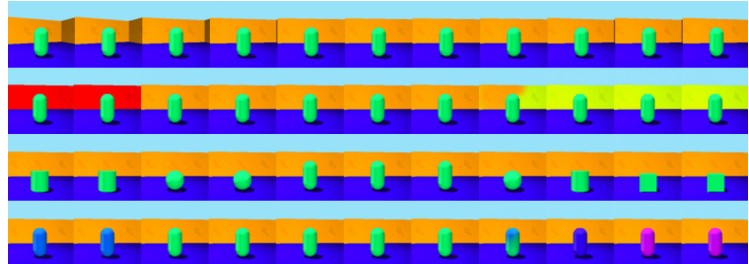

Figure 8: Latent variable visualization on Shapes3D with EncDiff + Dependency Sparsity. From top to bottom, the latent variables correspond to wall angle, wall color, object shape, and object color.

Penalty. Dependency sparsity gives the best results on both datasets, improving FactorVAE and DCI while maintaining backbone training stability.

**Qualitative evaluation.** A key goal of these experiments is to test whether dependency sparsity leads to more interpretable and disentangled latent representations in visual domains. Figure 8 shows latent traversals on Fashion (Xiao et al., 2017) with Flow. Individual latent coordinates correspond cleanly to gender, heel height, and upper width, with minimal interference across factors. The Shapes3D traversals in Figure 8 (EncDiff) show similarly sharp control, disentangling wall angle, wall color, object shape, and object color. These traversals illustrate that dependency sparsity yields latent axes that align with semantic attributes and preserve orthogonality among factors.

Figures 9, 10, and 11 further evaluate controllability via latent swapping. On Shapes3D, swapping a single factor cleanly transfers floor or wall color while leaving other factors intact. On Cars3D, EncDiff isolates azimuth and color. On MPI3D, rotation and background are controlled independently. These results highlight that dependency sparsity encourages *localized* and *non-overlapping* influences, enabling intuitive editing operations without unintended side effects. At the same time, the generative quality of the backbone (diffusion in this case) is preserved, with realistic outputs.

Together, these traversals and swaps reinforce the quantitative results: dependency sparsity not only improves disentanglement scores but also enhances interpretability and practical usability of the learned latents. By aligning latent dimensions with distinct semantic factors, it enables robust single-attribute manipulation and semantically meaningful latent arithmetic. These benefits are precisely what identifiability is meant to guarantee, providing further empirical validation of our theory.

**Additional results on controllable generation.** Moreover, we conduct further experiments on the benefit of recovering task-relevant representation for controllable generation. We consider the problems of

# D    DISCLOSURE STATEMENT

Grammar checks were performed using LLMs; no significant edits were made.

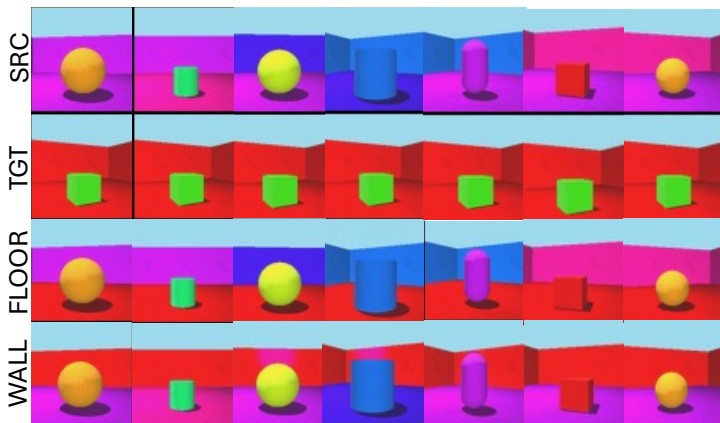

Figure 9: Latent variable visualization on Shapes3D with EncDiff + Dependency Sparsity. Top row: source. Second row: target. Each subsequent row modifies the source by swapping a single latent factor (floor color or wall color) from the target.

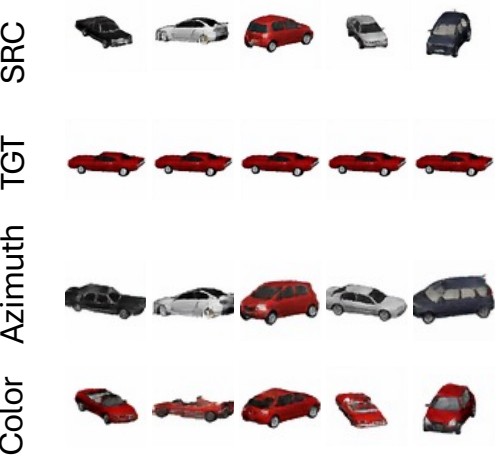

Figure 10: Latent variable visualization on Cars3D with EncDiff + Dependency Sparsity. Top row: source. Second row: target. Each subsequent row modifies the source by swapping a single latent factor (azimuth and color) from the target.

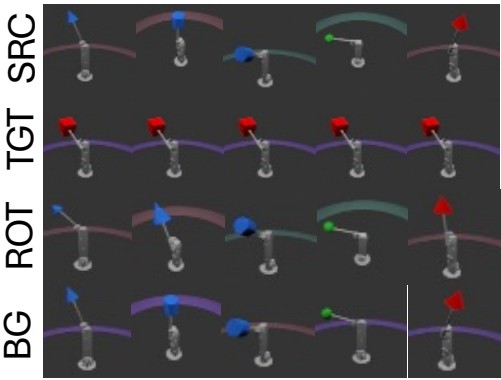

Figure 11: Latent variable visualization on MPI3D with EncDiff + Dependency Sparsity. Top row: source. Second row: target. Each subsequent row modifies the source by swapping a single latent factor (rotation and background) from the target.

