# OpenReview forum: "Diverse Dictionary Learning"
_ICLR.cc/2026/Conference — ICLR 2026 Poster_

### Official Review · Reviewer_P3Ud · 2025-10-27

**Soundness:** 3
**Presentation:** 2
**Contribution:** 2
**Rating:** 4
**Confidence:** 4

**Summary:**

This paper introduces the problem of diverse dictionary learning to make identifiability actionable in the real-world scenario. In particular, they study: in the general settings where full identifiability is unattainable, what can still be recovered with guarantees, and what biases could be universally adopted?

Section 3 introduces the main technical results.
The current presentation is hard to see it's impact in real problems.
There is a paragraph started with "Why does it matter in the real world?" By reading that paragraph, it's still not clear how the research problem in this paper relevant to any objective that matters in reality, such as improving the performance of a predictive model or dimension reduction.

The current simulations don't demonstrate the potential usefulness of the work. The synthetic data part focuses on Generalized Identifiability and Element Identifiability. Without knowing the possible impact of quantifying them, it is hard to see the potential value.

The section 4.2 is written in the way for readers who are familiar with the references. It does not provide sufficient explanation of the experimental results and their potential impact in practice.

Overall, it is hard to see the potential impact of this work. A better justification is needed.

**Strengths:**

The paper is free of grammatical errors.

**Weaknesses:**

See in the Summary; the later part.

**Questions:**

See in the Summary; especially in the later part.

---

> ### Author Response · Authors · 2025-11-22
> **We sincerely appreciate the time you devoted and the thoughtful comments (1/3)**
>
> Dear Reviewer P3Ud
>
> We sincerely appreciate the time you devoted and the thoughtful comments. In response, we have broadened several parts of the discussion to address your comments with care. Our detailed replies are provided below.
>
>
> **Q1**. More discussion on the potential impact of this work is needed.
>
>
> **A1**. Thanks a lot for your suggestions. We agree that discussing the broader impact of our work in more detail can be very helpful. In the revision, we have added a dedicated section (**Appendix B.7, L1489-1595**) highlighting potential implications across several domains. Below, we summarize the main points.
>
>
> In a nutshell, recovering the ground-truth data generative process is important for both predictive and non-predictive tasks. Machine learning is fundamentally a balance between inductive bias and data. By uncovering the hidden world underlying the data, we obtain principled, domain-agnostic inductive biases grounded in fundamental understanding rather than heuristics. These insights can be embedded directly into model architectures, training objectives, and evaluation protocols, yielding systems that are more robust, generalizable, and data-efficient.  Importantly, these inductive biases are not merely theoretical; they have already produced measurable improvements across diverse real-world domains.
>
> We first highlight two overarching benefits:
>
> - **Truthfulness**: It is well known that, given any pair of observationally equivalent models (e.g., models perfectly trained by MLE), without additional assumptions, there is no guarantee at all that these models actually recover the underlying data generative processes. This has led to the critical issue of non-identifiability of the nonlinear latent variable model (e.g., the classical work on the non-uniqueness of nonlinear ICA [1]), and has been widely validated by large-scale empirical studies on unsupervised learning (e.g., ICML19 best paper [2]). This raises significant challenges for tasks (e.g., transfer learning, controllable generation, compositional generalization, and mechanistic interpretability) where we need to understand the true process, rather than purely fitting the observed distribution. Therefore, understanding what aspects can still be recovered, and what inductive biases should be introduced to guide recovery, provides both theoretical guarantees and practical guidance on ensuring the truthfulness of the learning. These are exactly what our paper aims to address.
>
> - **Efficiency**: Recovering the underlying true generative factors provides a principled solution to improve the efficiency, since we only need to model those essential representations capturing what we are interested, instead of the whole high-dimensional space that also includes numerous irrelevant information or arbitrary noises. For instance, if we can make sure that the recovered latents will not be entangled with other latent variables, we can precisely only on those that relevant to our tasks (e.g., content for transfer learning, or specific latent concepts we need to modify) instead of the whole latent spaces, not only reducing the cost but enforcing the model to focus only on those relevant, improve the efficiency in a principled way.

---

> ### Author Response · Authors · 2025-11-22
> **We sincerely appreciate the time you devoted and the thoughtful comments (2/3)**
>
> Our simulation rigorously validates the established identifiability theory, and our visual experiments (Section 4.2) further illustrate the practical implications of the corresponding inductive bias in practice. Across datasets and backbone models, the results consistently show that adding the identifiability-guided regularization (dependency sparsity) enables the model to recover the underlying generative factors. This not only improves disentanglement performance, which is itself practically important, but also directly benefits the following application scenarios:
>
> - **Mechanistic Interpretability**:  In many scenarios, we aim to study the underlying concepts that generate the responses of LLMs, for many reasons such as investigating the root of hallucination or explaining the mechanisms for specific responses or behaviors. This is related to the field of Mechanistic Interpretability, where the Sparse Autoencoder (SAE) has been popular. However, as known by the community [3], SAE is based on the theory of Sparse Dictionary Learning, and there are some open problems:
>
>    - First, SAE can only deal with linear data due to the lack of nonlinear identifiability of sparse dictionary learning. However, the real world is full of nonlinearity. Assuming everything is linear simplifies the procedure and will unavoidably bring bias and cannot fully recover the truth.
>
>    - Moreover, SAE encourages sparsity over the latent variables. As a result, it needs an extremely high-dimensional vector to capture real-world concepts (e.g., millions of dimensions), and some features are easy to be absorbed due to its encouragement of latent sparsity.
>
>    In contrast, our Diverse Dictionary Learning encourages the sparsity of the Jacobian (i.e., dependency sparsity) instead of latent diversity, and can handle nonlinear generative processes with identifiability guarantees. This provides a principled solution to both open problems of SAE, which is highly significant for the whole field of mechanistic interpretability.
>
>    **Connection to our experiments**: As shown in our experiments in Section 4.2, enforcing dependency sparsity appears more effective than the latent sparsity used in traditional SAEs. Moreover, as noted in our paper (L465), a recent concurrent work demonstrates strong empirical performance from regularizing dependency sparsity in mechanistic interpretability [4], offering direct empirical support for our theory on interpreting large-scale language models.
>
> - **Transfer Learning**: In transfer learning, it is important to disentangle the invariant and changing parts, such as disentangling invariant content from changing styles. Identifiability has been widely leveraged in the previous literature on guaranteeing reliable and efficient domain adaptation (e.g., [5,6,7]). Given two observed variables, our generalized identifiability results guarantee the recovery of the shared and private parts of their latent variables, under a simple regularization of dependency sparsity. This provides a flexible framework for transfer learning that can really capture the essential part across domains.
>
>    **Connection to our experiments**: Both our synthetic and image experiments show that incorporating dependency sparsity recovers the underlying generative concepts even in the general nonparametric setting. This forms a solid basis for identifying content and style variables in transfer learning, since it ensures that the estimated content and style latents align with their true counterparts without hidden entanglement that would otherwise disrupt transfer.
>
> - **Controllable Generation**: A perfect prediction machine can only master mining correlations, while leveraging the true causation relies on identifiability. For instance, if we want to add eyeglasses on a kid’s face, due to the overwhelming correlation between wearing eyeglasses and the relatively higher age, models sometimes will also increase the age of the kid, even for large foundational models. This negligence of the true underlying process is one of the fundamental reasons why large models, although extensively trained on web-scale data, can still provide many extra changes that are out of our control. Previous work has already shown that, by incorporating inductive bias guided via identifiability, models can achieve precise control on the generated images [8, 9].
>
>    **Connection to our experiments**: Our nonlinear disentanglement results show that dependency sparsity prevents unintended mixing between latent factors. This is precisely the issue that leads to unwanted edits in controllable generation, where modifying one attribute (such as adding eyeglasses) can unintentionally alter others (such as age). By keeping each latent factor isolated from irrelevant ones, the model learns representations where edits stay local and predictable. These observations directly support the type of structure needed for reliable controllable generation.

---

> > ### Author Response · Authors · 2025-11-22
> > **We sincerely appreciate the time you devoted and the thoughtful comments (3/3)**
> >
> > Similarly, identifiability theory, validated rigorously by our simulations and further supported by our visual experiments, extends to many other practical applications. To name only a few:
> >
> > - **Multi-modal Alignment**: Similarly, one may consider multiple modalities as multiple sets of observed variables, where the semantically meaningful concepts are those that are shared by multiple modalities, and the modality-specific concepts (e.g., texture for images, volumes for audio) are those private latent variables. By formalizing the process as a general dictionary learning problem, many previous works have already shown the practical implications of identifiability, such as addressing information misalignment [10] and capturing complex interactions across different modalities [11].
> >
> > - **Scientific Discovery**: Understanding the hidden truth from observation is always one of the main tasks of scientific discovery. Newton observed the falling apple (observation $X$), then he got curious, studied much, and found out it was actually gravity (latent variables $Z$) causing items to fall. Therefore, the fundamental task of scientific discovery may be summarized into the simple equation of $X=f(Z)$, where we aim to recover $Z$ based solely on $X$, which is exactly our task of general dictionary learning. Of course, the guarantee that the recovered latent variables $\hat{Z}$ correspond to the ground-truth latents $Z$ in a meaningful way is what matters the most, and this is exactly the focus of our generalized identifiability theory. There have been numerous successful applications of identifiable representation learning in the literature, such as dynamic systems like climate change [12], robotics [13], neural imaging [14], and genomics [15].
> >
> >
> > Of course, the list is non-exclusive, and will only grow faster given the development of large-scale models. The reason is simple: even with infinite data and computation, without identifiability, models may achieve perfect predictions but cannot be guaranteed to recover the underlying truth. Thus, the closer we are to that boundary, the more efforts we should put into going beyond the pure correlation. We sincerely appreciate your suggestions on expanding the discussions on potential applications, and we believe these added examples shed light on the necessity of understanding and leveraging the hidden world.
> >
> >
> >
> >
> > [1] Nonlinear Independent Component Analysis: Existence and Uniqueness Results. Neural Networks, 1999
> >
> > [2] Challenging Common Sssumptions in the Unsupervised Learning of Disentangled Representations. ICML 2019.
> >
> > [3] Open Problems in Mechanistic Interpretability. arXiv, 2025
> >
> > [4] Jacobian sparse autoencoders: Sparsify computations, not just activation. ICML 2025
> >
> > [5] Self-Supervised Learning with Data Augmentations Provably Isolates Content from Style. NeurIPS 2021
> >
> > [6] Partial Identifiability for Domain Adaptation. ICML 2022
> >
> > [7] Subspace Identification for Multi-source Domain Adaptation. NeurIPS 2023
> >
> > [8] Learning Vision and Language Concepts for Controllable Image Generation. ICML 2025
> >
> > [9] Multi-domain Image Generation and Translation with Identifiability Guarantees. ICLR 2023
> >
> > [10] SmartCLIP: Modular Vision-language Alignment with Identification Guarantees. CVPR 2025
> >
> > [11] Causal Representation Learning From Multi-modal Biomedical Observations. ICLR 2024
> >
> > [12] Marrying Causal Representation Learning with Dynamical Systems for Science. NeurIPS 2024
> >
> > [13] BISCUIT: Causal Representation Learning from Binary Interaction. UAI 2023
> >
> > [14] Unsupervised Feature Extraction by Time-Contrastive Learning and Nonlinear ICA. NeurIPS 2016
> >
> > [15] Causal Representation Learning Made Identifiable by Grouping of Observational Variables. ICML 2024

---

> > > ### Author Response · Authors · 2025-12-02
> > > **Summary of the rebuttal**
> > >
> > > Dear AC,
> > >
> > > Thanks so much for your time and effort. For convenience, we would like to provide a brief summarization of the rebuttal with Reviewer P3Ud.
> > >
> > > The reviewer’s main question concerned the potential impact of the work. In response, we added a dedicated section outlining implications across several domains. Specifically, we highlighted **two overarching benefits** and **five concrete application areas** that directly benefit from our results, supported by clear methodology and representative references. We believe the **question has been fully addressed** and that the **rating should be adjusted accordingly**.
> > >
> > > Sincerely,
> > >
> > > Authors of Submission 564

---

### Official Review · Reviewer_U1DX · 2025-10-30

**Soundness:** 3
**Presentation:** 3
**Contribution:** 2
**Rating:** 6
**Confidence:** 3

**Summary:**

The paper studies nonlinear dictionary learning under a general generative model $X = g(Z)$, where both the latent variables $Z$ and the generator $g$ are unknown. It introduces a set-theoretic notion of identifiability, focusing on recoverability of set operations: intersections, complements, and symmetric differences. Under assumptions of positive density and dependency sparsity, the authors show that these structural dependencies are identifiable, and that element-wise identifiability follows when the system exhibits “sufficient diversity.”

**Strengths:**

1) The paper provides a practical and general inductive bias that improves disentanglement without model-specific changes, May be useful for interpretability and modular representation learning.
2) Strong theoretical results: generalized and structural identifiability (Theorems 1–3) under mild assumptions.
3) The “sufficient diversity” condition unifies and relaxes previous sparsity-based identifiability assumptions.
4) Introduces a novel set-theoretic identifiability framework that generalizes traditional recovery notions through operations on supports of $D_z g$.

**Weaknesses:**

Below I list a few weaknesses of the paper:

1. The “sufficient nonlinearity” and independance conditions on $D_z g$ are elegant but may be hard to verify or hold only asymptotically. Some discussion of finite-sample behavior or diagnostics would help.

2. Theoretical results assume $\ell_0$ regularization on $|D_z \hat g|_0$, but experiments use an $\ell_1$ proxy. A clearer justification or sensitivity analysis for this relaxation would strengthen the bridge between theory and practice.

3. Penalizing full Jacobians can be expensive for large decoders or diffusion models. Runtime or memory comparisons against latent-sparsity or Hessian-based baselines are missing.

4. Theoretical claims focus on structural identifiability, but evaluations (e.g., DCI, FactorVAE) measure disentanglement indirectly. A metric targeting recovery of the Jacobian’s support would better reflect the paper’s main claims.

5. Noise and model misspecification (e.g., non-additive noise, partial non-invertibility) are briefly mentioned but not deeply explored; results may not generalize to such cases.

**On another note, I am not sure how good of  a fit is this theoretical endeavor suitable for ICLR**

**Questions:**

Do you have any results or insights on how accurately the support of $D_z g$ can be recovered with finite samples or under noise?
Can the diversity condition be checked empirically?
Would it be possible to report direct recovery metrics?

---

> ### Author Response · Authors · 2025-11-22
> **We are very grateful for your insightful and constructive feedback (1/3)**
>
> Dear Reviewer U1DX,
>
> We are very grateful for your insightful and constructive feedback. We are glad that you found the inductive bias helpful, the theory strong, and the framework novel. Following your suggestions, we have added **new experiments** and **expanded discussions** in the updated manuscript. Please find the detailed responses as follows:
>
> ---
>
> **Q1**.The “sufficient nonlinearity” and independence conditions on $D_z g$ are elegant but may be hard to verify or hold only asymptotically. Some discussion of finite-sample behavior or diagnostics would help
>
> **A1**. Thank you very much for this thoughtful suggestion. The sufficient nonlinearity assumption is meant to ensure that the Jacobian varies enough across samples so that its Jacobian vectors span the relevant support. Although this condition may seem demanding at first glance, it is usually quite mild in practice. For each observed variable $X_i$, the requirement involves only $|(D_Z g)\_{i,\cdot}|_0$ samples, which is simply the number of latent variables that influence $X_i$, and this number is typically far smaller than the available sample size.
>
> When $g$ is smooth and the latent distribution has a continuous density, Jacobian evaluations at independently drawn samples form continuous random vectors. Such vectors are in general position with probability one, which means that a small number of random samples already produces Jacobian rows that span the required support. For example, if $X_i$ depends on five latent coordinates, then roughly five random samples are usually sufficient, even in much higher-dimensional systems.
>
> We appreciate the chance to clarify this point, and we have added a detailed discussion in the updated manuscript (**Appendix B.4, L1428-1443**) to help readers see why this assumption is typically easy to satisfy in finite sample settings.
>
> ---
>
> **Q2**. Theoretical results assume $\ell_0$ regularization on $\|D_{\hat{Z}} \hat{g} \|_0$, but experiments use an $\ell_1$ proxy. A clearer justification or sensitivity analysis for this relaxation would strengthen the bridge between theory and practice.
>
> **A2**. Thanks for raising this important point. For the regularization term, directly utilizing the $\ell_0$ penalty may be computationally infeasible as it results in a discrete optimization problem. To overcome this issue, we adopt the $\ell_1$ regularizer, which has been extensively studied in the literature as a gradient-friendly version of $\ell_0$ regularization [1, 2]. In the context of nonparametric identifiability, using $\ell_1$ as a sparsity regularization is also the standard strategy [3, 4].
>
> Interestingly, Zheng and Zhang [3] had conducted experiments across different versions of the regularization, basically different hybrids of $\ell_0$ and $\ell_1$ (smoothly clipped absolute deviation (SCAD) penalty, minimax concave penalty (MCP), and $\ell_1$ penalty). From the Figure 8 in [3], we can see that different variants do not differ much, and thus we choose the simplest $\ell_1$ regularization.
>
>
>
> [1] Sharp thresholds for high-dimensional and noisy sparsity recovery using $\ell_1$-constrained quadratic programming (Lasso), IEEE Transactions on Information Theory, 2009
>
>
> [2] Model selection in Gaussian graphical models: High-dimensional consistency of $\ell_1$-regularized MLE. NeurIPS 2008
>
> [3] Generalizing nonlinear ICA beyond structural sparsity. NeurIPS 2023
>
> [4] Synergy between sufficient changes and sparse mixing procedure for disentangled representation learning. ICLR 2025

---

> ### Author Response · Authors · 2025-11-22
> **We are very grateful for your insightful and constructive feedback (2/3)**
>
> **Q3**. Penalizing full Jacobians can be expensive for large decoders or diffusion models. Runtime or memory comparisons against latent-sparsity or Hessian-based baselines are missing.
>
>
> **A3**. Thank you for the insightful question. We fully agree that computing the full Jacobian of a large model can be expensive. Fortunately, recent work [1] shows that dependency sparsity regularization remains practical even at scale up to common LLMs when combined with two standard strategies.
>
> - **Apply latent sparsity first**.  Large models often have high-dimensional latent spaces, but for any given input, many latent coordinates are inactive or irrelevant. A common approach is to first identify the active coordinates and compute the Jacobian only with respect to this subset. Since the active block is often tiny compared to the full latent space, this reduces both computation and memory by several orders of magnitude in typical transformer architectures [1]. In practice, the Jacobian is rarely formed as a dense $d_x \times d_z$ matrix, but only as a restricted slice that corresponds to the active latent directions for that specific input.
>
> - **Use efficient closed-form expressions**. For several widely used architectures, the Jacobian with respect to selected latent directions has efficient factorizations. As shown in [1], models with residual attention and feedforward structure admit closed-form expressions for the relevant Jacobian blocks that require only a few matrix multiplications and inexpensive elementwise operations. This avoids repeated backward passes through the full model and keeps the cost manageable even when the underlying network is large.
>
> With this approach, training a large model with our dependency sparsity regularization is reported to be only about twice as slow as training with standard $\ell_1$ regularization on the latent variables $Z$ [1]. In light of your helpful suggestion, we have added the corresponding discussion in the updated manuscript (**Appendix B.5, L1445-1465**), and we believe it will be useful for practitioners working with large architectures.
>
> [1] Jacobian sparse autoencoders: Sparsify computations, not just activations. ICML 2025
>
> ---
>
>
> **Q4**. Theoretical claims focus on structural identifiability, but evaluations (e.g., DCI, FactorVAE) measure disentanglement indirectly. A metric targeting recovery of the Jacobian’s support would better reflect the paper’s main claims.
>
>
> **A4**. Thank you for the question. To clarify, our core theoretical claims concern the identifiability of latent variables themselves, as captured by Theorem 1, Proposition 1, and Theorem 3. Structure identifiability in Theorem 2 is an additional result for cases where the mixing structure is of interest. In Definition 6 and Definition 8, identifiability is defined through whether the recovered variables avoid being functions of irrelevant ones. This is the notion that standard disentanglement metrics aim to test. The evaluations we provide therefore target the central theoretical claims.
>
> That said, we agree that a metric focused on recovering the support of the Jacobian would provide a clear view of the structural aspect. Since Theorem 2 has independent value for structure learning, we ran new experiments on randomly generated structures to evaluate structure recovery directly. The setting follows our simulation in the paper. We report the structural hamming distance (SHD) between the estimated and ground-truth structures. The results are as follows:
>
> | Number of variables | SHD |
> |----------|-----|
> | 3 | 0 |
> | 4 | 0 |
> | 5 | 0.66 ± 0.94 |
> | 6 | 1.33 ± 0.94 |
>
>
> These results show that the Jacobian support can indeed be recovered, consistent with Theorem 2.

---

> > ### Author Response · Authors · 2025-11-22
> > **We are very grateful for your insightful and constructive feedback (3/3)**
> >
> > **Q5**. Noise and model misspecification (e.g., non-additive noise, partial non-invertibility) are briefly mentioned but not deeply explored; results may not generalize to such cases.
> >
> >
> > **A5**. Thanks for raising this. For noise and invertibility, we follow the standard setup in the identifiability literature because violations of these basic conditions usually require much stronger assumptions to compensate. That said, we agree that more discussion on how to handle these cases is helpful. We have added the following discussion in the updated manuscript as a separate subsection (**Appendix B.6, L1467-1489**):
> >
> > - **Handling general noises**: Most identifiability results for latent variable models focus on additive independent noise or noiseless settings. This holds for both classical linear models [1] and recent nonparametric work [2]. The main difficulty is separating noise from latent variables, since in the general form, they can be entangled in complex ways. Recent work [3], based on the Hu Schennach theorem [4], shows that general noise can be separated under additional assumptions on the generative function $f$ and on conditional independence across groups of observed variables. As noted in Remark 1, under the same conditions, our results extend to settings with general noise.
> >
> > - **Handling partial non-invertibility**: Invertibility and its variants are among the most common assumptions in the literature. Without additional assumptions, it can even be necessary, since information that is lost during generation cannot be recovered in principle. In scenarios where partial non-invertibility must be addressed, one may consider incorporating temporal information together with sufficient changes in the nonstationary transition [5], which can provide the extra information needed for recovery. Since our theory focuses on the general setting without any auxiliary information, we adopt the standard assumption of invertibility.
> >
> > We hope these discussions will be helpful for applications in the corresponding scenarios. Please feel free to let us know if there are any further questions.
> >
> > [1] Identifiability of a linear relation between variables which are subject to error. Econometrica: Journal of the Econometric Society, 1950.
> >
> > [2] Identifiability of latent-variable and structural-equation models: from linear to nonlinear. Annals of the Institute of Statistical Mathematics, 2024
> >
> > [3] Nonparametric factor analysis and beyond, AISTATS 2025
> >
> > [4] Instrumental variable treatment of nonclassical measurement error models. Econometrica, 2008
> >
> > [5] CaRiNG: Learning temporal causal representation under non-invertible generation process, ICML 2024
> >
> > ---
> >
> > **Q6**. On another note, I am not sure how good of a fit is this theoretical endeavor suitable for ICLR.
> >
> >
> > **A6**. Thanks for the question. According to the ICLR 2026 call for papers, the venue welcomes work on representation learning. Our paper contributes to this area by providing a theoretical foundation for the identifiability of latent representations and by introducing a practical inductive bias that improves the recovery of these representations in existing models. In addition, many previous works on identifiable representation learning have appeared at ICLR (e.g., [1,2,3,4,5]), showing that this research direction aligns well with the conference. For these reasons, we believe our work is a good fit for ICLR.
> >
> >
> > [1] Multi-view causal representation learning with partial observability, ICLR 2024
> >
> > [2] Synergy between sufficient changes and sparse mixing procedure for disentangled representation learning, ICLR 2025
> >
> > [3] Identifiable latent polynomial causal models through the lens of change, ICLR 2024
> >
> > [4] Identifiable exchangeable mechanisms for causal structure and representation learning, ICLR 2025
> >
> > [5] Content-style learning from unaligned domains: Identifiability under unknown latent dimensions, ICLR 2025
> >
> >
> > ---
> >
> >
> > **Q7**. Do you have any results or insights on how accurately the support of $D_z g$ can be recovered with finite samples or under noise? Can the diversity condition be checked empirically? Would it be possible to report direct recovery metrics?
> >
> > **A7**. Thanks so much for these insightful questions. Yes, we can accurately recover the support of $D_z g$ in practice, as detailed in A4. We also tried adding Gaussian noise, and the results are similar. Moreover, we believe it is empirically feasible to check the structural diversity condition by just looking at the recovered structure, since Theorem 2 only requires a sparsity regularization and does not put any constraints on the ground-truth structure. The direct recovery metrics (SHDs) are as follows:
> >
> > | Number of variables | SHD |
> > |----------|-----|
> > | 3 | 0 |
> > | 4 | 0 |
> > | 5 | 0.66 ± 0.94 |
> > | 6 | 1.33 ± 0.94 |

---

> > > ### Author Response · Authors · 2025-12-02
> > > **Summary of the rebuttal**
> > >
> > > Dear AC,
> > >
> > > Thanks so much for your time and effort. For convenience, we would like to provide a brief summarization of the rebuttal with Reviewer U1DX.
> > >
> > > We are glad the reviewer found the inductive bias helpful, the theory strong, and the framework novel. Their questions focused on clarifying several specific details in the manuscript, not on the originality or soundness of the work. We have provided detailed responses to each point, supported by new experiments and further discussion. We believe all questions have been fully resolved with clear evidence and comprehensive explanations.
> > >
> > > Sincerely,
> > >
> > > Authors of Submission 564

---

### Official Review · Reviewer_KUfT · 2025-10-31

**Soundness:** 3
**Presentation:** 4
**Contribution:** 3
**Rating:** 6
**Confidence:** 2

**Summary:**

The authors introduce a new, weaker form of identifiability based on "set-theoretic indeterminacy". The main theoretical result is the Set-Theoretic Guarantees (Theorem 1): that even when full recovery is impossible, structural relationships between latent variables are identifiable. Specifically, the intersection ($I_K \cap I_V$, shared factors), complement ($I_K \setminus I_V$, exclusive factors), and symmetric difference ($I_K \Delta I_V$, unique factors) of latent variable sets are shown to be disentangled.

These identifiability guarantees (Theorems 1 & 2) are achieved by introducing a "simple inductive bias" during estimation: a sparsity regularization on the dependency structure (i.e., the Jacobian $D_{\hat{Z}}\hat{g}$). This is a practical regularization rather than a strict assumption about the data-generating process itself.

The authors show that their set-theoretic guarantees naturally extend to full, element-wise identifiability (the traditional goal) if a "Sufficient Diversity" condition (Assumption 2) is met. This condition, which is shown to be a weaker and more general structural requirement than in prior work, essentially ensures the dependency structure is rich enough for each latent variable to be isolated in its own "atomic region" of the set-theoretic Venn diagram.

Finally, the authors conduct experiments on synthetic and real-world image datasets to validate their theory and demonstrate the practical benefits of the proposed dependency sparsity bias.

**Strengths:**

- The research area of expanding traditional dictionary learning inductive priors to learn true latents of the data is interesting and a valuable area of research.
- The paper appears theoretically sounds. The authors provide rigorous definitions for all new theoretical concepts and include detailed proofs in the appendix. I took a brief look at the proofs and did not find any issues, although I was not able to follow everything and there is a good chance I could have missed an issue.
- The writing is very clean and easy to understand (although the math is dense) with associated well made figures backing up the intuition with toy examples which really helped my understanding of the paper.

**Weaknesses:**

- The paper discusses SAEs in the introduction but never again throughout the paper. I was expecting some experiments or discussion about whether Diverse Dictionary Learning could be used in-place of traditional SAE losses. This shouldn't be difficult to do as there are many small models + SAE libraries available and would be really interesting to compare. Indeed, SAEs assume the linear representation hypothesis, but there is some decent evidence that the LRH is at least somewhat true in many scenarios - how does the proposed method do when examining model internals?
- The main theory relies heavily on Assumption 1 (Sufficient Nonlinearity). I checked the referenced papers in Lines 303 and found that some of them contained similar statements but it was hard to determine if they were identical. I think the paper would be much stronger if the authors provided a bit more intuition as to why Assumption 1 is reasonable to make. Even better, if there were some small experiments to show that this generally holds empirically, that would strengthen the paper a lot. However, if this really is a very standard assumption, then this evidence is not wholly necessary, but I would appreciate the authors highlighting exactly which equations in which papers make the same assumption so I can more readily compare the two. (For example, I am guessing Assumption 1 in Lachapelle et al., 2022 makes the same argument?)

**Questions:**

- The conclusion ends suggesting to explore identifiability in foundation models. How would one practically implement and optimize this dependency sparsity bias in a large-scale transformer as the full Jacobian of a transformer's activations is computationally intractable.
- How does the proposed diverse dictionary learning compare to traditional SAE methods for *interpreting* model architectures rather than solely looking at traditional datasets? Would the result be identical to that of Farnik et al. 2025 (Jacobian Sparse Autoencoders)?
- I am not sure I fully understand how the terminology and figures would map onto a real world example. In a complex domain like images, what would these "atomic regions" practically represent? For example, if $I_1$ = "dog" latents and $I_2$ = "cat" latents, could the intersection $I_1 \cap I_2$ be the latent for "furry," and the complement $I_1 \setminus I_2$ the latent for "barks"? How does this set-based view compare to traditional feature representations?

Generally, I do not see many issues with the paper, and if my confusions are addressed during the rebuttal I would be happy to raise my score accordingly!

---

> ### Author Response · Authors · 2025-11-22
> **We truly appreciate your thoughtful and constructive comments (1/3)**
>
> Dear Reviewer KUfT,
>
> We truly appreciate your thoughtful and constructive comments. We are glad that you found the problem interesting, the results sound, and the writing easy to understand. In light of the great suggestions, we have added **new experiments** and **expanded discussions** in the updated manuscript. Please see our detailed point-by-point responses below.
>
> ---
>
> **Q1**. More discussion on the relation between SAEs and diverse dictionary learning, especially on exploring the inner representation of models.
>
> **A1**. This is a very great suggestion, thanks a lot! This is indeed one of our main motivations, since diverse dictionary learning directly generalizes linear dictionary learning, and thus the corresponding regularization of dependency sparsity should also have some benefits compared to the traditional latent sparsity of SAE. In light of your great suggestion, we have added detailed discussions in a dedicated subsection (**Appendix B.3, L1382-1426**) to dive deeper into this topic, which we summarized as follows:
>
> - **Discussion**:  Mechanistic interpretability often seeks to uncover the underlying concepts that drive the behavior of large language models, whether to understand sources of hallucination or to explain model responses. Sparse Autoencoders (SAEs) have been widely used for this purpose, but, as noted in the community [3], SAEs rely on sparse dictionary learning and therefore inherit its limitations:
>
>    - First, SAEs fundamentally assume a **linear generative function rather than only linear relations in LRH**, since sparse dictionary learning lacks nonlinear identifiability. As you mentioned, LRH is a reasonable working assumption in many contexts, but LRH concerns the linear relation within the latent space of $Z$ rather than the linearity of the generative map $g$ in $X=g(Z)$. In reality, most models have highly nonlinear generative functions (for example, due to nonlinear activations such as ReLU or GeLU). Assuming linearity at the generative level simplifies implementation but inevitably introduces bias and prevents full recovery of the true latent factors in these nonlinear settings.
>
>       Different from the theoretical foundation of SAEs (i.e., sparse dictionary learning), our diverse dictionary learning provides theoretical guarantees for nonlinear latent variable models, offering a provably rigorous tool for mechanistic interpretability in almost all real-world scenarios, covering the previously unsupported nonlinear cases.
>
>    - Second, SAEs impose **sparsity directly on the latent variables**. This often forces extremely high-dimensional latent spaces (e.g., millions of units) to represent real-world concepts. More, latent sparsity can cause feature splitting and absorption, where important concepts are lost due to the encouragement of sparse latents. For instance, if we only maximize the latent sparsity, each feature will only corresponds to several samples, capture concepts that are very specific (e.g., persian cats) versus more general concepts (e.g., cats).
>
>       In contrast, diverse dictionary learning encourages dependency sparsity (Jacobian sparsity) rather than latent sparsity, and is designed for nonlinear generative models with identifiability guarantees. This provides a principled solution to both limitations of SAEs and has meaningful implications for mechanistic interpretability.
>
> - **Empirical Evidence**: As you noted, the recent Jacobian Sparse Autoencoder (JSAE) work [1] provides extensive empirical support for the same dependency sparsity regularization required by our diverse dictionary learning theory. Their results suggest that replacing traditional SAE losses with dependency sparsity–based losses can improve both interpretability and efficiency. We therefore refer readers in mechanistic interpretability to their thorough empirical study (see L465).
>
>    At the same time, there is one perspective that [1] has not evaluated against other SAE baselines, i.e., the number of dead features. Thus, we conducted new experiments on the OpenWebtext with GPT2-Small, comparing JSAE with Top-K SAE [2] and Batch Top-K SAE [3], with the latent dimension as $12,288$. These results, together with the comprehensive experiments in [1], further support the advantages of dependency sparsity.
>
>    |Method|Number of Dead Features|
>    |-|-|
>    |Top-K SAE|439|
>    |Batch Top-K SAE|207|
>    |JSAE|62|
>
> [1] Jacobian sparse autoencoders: Sparsify computations, not just activations. ICML 2025
>
> [2] Scaling and evaluating sparse autoencoders. ICLR 2025
>
> [3] BatchTopK sparse autoencoders. NeurIPS 2024 Workshop

---

> ### Author Response · Authors · 2025-11-22
> **We truly appreciate your thoughtful and constructive comments (2/3)**
>
> **Q2**. More discussion on Assumption 1 (Sufficient Nonlinearity), and highlight it in previous papers.
>
> **A2**. Thank you a lot for the suggestions. For exact references, the following assumptions serve as direct examples:
>
> - Assumption ii of Theorem 1 in [1];
>
> - Assumption i of Theorem 3.1 in [2];
>
> - Assumption 3.1 (ii) in [3];
>
> - Assumption 1 (ii) in [4].
>
> To make the intuition of that assumption clearer, we have added a more detailed discussion (**Appendix B.4, L1428-1443**), inspired by these earlier works. We summarized it as follows:
>
> > The assumption ensures that the Jacobian can capture the connective structure in the nonlinear case. It rules out pathological situations where all samples come from overly restricted sub populations that only span a degenerate subspace. The first part guarantees that there exist enough data points (perhaps just a few) for the Jacobian to span the relevant support, which is almost always satisfied asymptotically. It is also worth noting that identifiability, the goal of the theorem itself, is an asymptotic notion. The second part $\operatorname{supp}((D_Z g(z^{(k)}) H)\_{i,\cdot}) \subseteq \operatorname{supp}((D\_{\hat{Z}}\hat{g})\_{i,\cdot}).$ is also mild. We have $(D\_{\hat{Z}} \hat{g} (\hat{z}))\_{i,\cdot} = (D_Z g (z))\_{i,\cdot} h(z, \hat{z})$, which already lies inside $\operatorname{supp}((D\_{\hat{Z}}\hat{g})\_{i,\cdot})$. Even in rare cases where a specific matrix fails to match the support due to particular value combinations, the condition still holds asymptotically, since it only requires the existence of one matrix in the space that is not ill-posed.
>
> Thanks again for your thoughtful feedback. Please feel free to let us know if anything is still unclear.
>
>
> [1] On the identifiability of nonlinear ICA: Sparsity and beyond. NeurIPS 2022
>
> [2] Generalizing nonlinear ICA beyond structural sparsity. NeurIPS 2023
>
> [3] Identification of nonlinear latent hierarchical models. NeurIPS 2023
>
> [4] Counterfactual generation with identifiability guarantees. NeurIPS 2023
>
> ---
>
> **Q3**. How would one practically implement and optimize this dependency sparsity bias in a large-scale transformer, as the full Jacobian of a transformer's activations is computationally intractable?
>
> **A3**. Thank you for the insightful question. We fully agree that computing the full Jacobian of a large transformer layer is intractable. Fortunately, as demonstrated in recent work [1], there are practical strategies that make dependency–sparsity regularization feasible at scale:
>
> - **Only compute the active block.**  A first step is to apply standard latent sparsity to filter out inactive latent variables, which are typically numerous in large models. For each token, only a relatively small number of latents are active, and the Jacobian needs to be computed only with respect to these coordinates. This drastically reduces the computation. As reported in [1], this simple preprocessing can reduce the effective Jacobian size by roughly six orders of magnitude in typical transformer settings.
>
> - **Use an efficient closed form.** In addition, for certain architectures such as GPT-2–style models, Farnik et al. [1] derive an explicit closed-form expression for the relevant Jacobian block. This formula requires only three matrix multiplications and a few pointwise operations, rather than multiple backward passes through the entire network. With this approach, training a Jacobian-regularized model is reported to be only about twice as slow as training a standard SAE.
>
>
> [1] Jacobian sparse autoencoders: sparsify computations, not just activations. ICML 2025

---

> ### Author Response · Authors · 2025-11-22
> **We truly appreciate your thoughtful and constructive comments (3/3)**
>
> **Q4**. I am not sure I fully understand how the terminology and figures would map onto a real world example. In a complex domain like images, what would these "atomic regions" practically represent? For example, if $I_1$ = "dog" latents and $I_2$ = "cat" latents, could the intersection $I_1 \cap I_2$ be the latent for "furry," and the complement $I_1 \setminus I_2$ the latent for "barks"? How does this set-based view compare to traditional feature representations?
>
> **A4**. Yes, your understanding is exactly right. If $I_1$ represents "dog'' latents and $I_2$ represents "cat'' latents, then the intersection $I_1 \cap I_2$ corresponds to latent factors shared by both classes, such as "furry.'' Likewise, the complement $I_1 \setminus I_2$ represents factors unique to dogs, such as "barks.''
>
> This is precisely why we find the set-theoretic view appealing. Instead of trying to recover every individual latent variable, our framework focuses on identifying the *shared* and *private* components across different groups. In many applications, this is exactly the level of recovery one needs. For example, transfer learning often relies on preserving the shared content (the intersection) while allowing style or class-specific variations (the complements) to change.
>
> Conceptually, this set-based perspective is compatible with traditional feature representations, but our contribution is to provide *provable guarantees* for recovering these informative sets in the general nonparametric setting. This captures the structure most downstream tasks rely on, without requiring full element-wise disentanglement, which typically demands additional assumptions.
>
> Thanks again for the time and care you invested in reviewing our work. Your suggestions are really helpful. If any part of our responses remains unclear, please feel free to let us know, and we will be glad to clarify further.

---

> > ### Comment · Reviewer_KUfT · 2025-11-26
> >
> > Thank you to the authors for the detailed response! You have clarified my confusions well and I think the additional discussion points and experiments on the jacobian SAE have made the paper stronger.
> >
> > Overall, I think the paper is a positive, especially for interpretability researchers to get a better theoretical justification for various sparsity constraints. Albiet, I am a bit fuzzy on some of the math presented and am not familiar with older identifiability and sparsity literature so I am not 100% confident in my assessment of the contributions and claims. That being said, I am happy to tentatively raise my score to 8 for the time being but I am also interested to see the other reviewers engage in the discussion process which could help confirm my thinking and final score.

---

> > > ### Author Response · Authors · 2025-11-26
> > > **Thanks so much for your encouragement**
> > >
> > > Thanks very much for your encouragement. Your suggestions are constructive and truly appreciated. We are glad that our responses are helpful. Regarding the theoretical contributions and conditions, we have added a new subsection in the updated manuscript (Appendix B.2., L1350 to 1381). We are familiar with these lines of theoretical work and believe our theory provides a distinctive perspective by showing what can still be recovered when full identifiability is unattainable in the general case, as well as what forms of inductive bias may be broadly beneficial. Please feel free to reach out anytime if you have any further questions. We would be very glad to follow up.

---

> > > > ### Author Response · Authors · 2025-12-02
> > > > **Summary of the rebuttal**
> > > >
> > > > Dear AC,
> > > >
> > > > Thank you for your time and effort. For convenience, we provide a brief summary of our rebuttal to Reviewer KUfT.
> > > >
> > > > We are glad the reviewer found the problem interesting, the results sound, and the writing easy to follow. We have provided detailed responses to all questions, and the reviewer confirmed that the clarifications, additional discussion, and further experiments strengthened the paper. As a result, they have **raised their score from 6 to 8**.
> > > >
> > > > Sincerely,
> > > >
> > > > Authors of Submission 564

---

### Official Review · Reviewer_xMy9 · 2025-11-02

**Soundness:** 2
**Presentation:** 2
**Contribution:** 3
**Rating:** 2
**Confidence:** 3

**Summary:**

This paper studies a set theoretic notion of dictionary learning, where the properties of the collection of atoms and the model are defined at the full set theoretic level of abstraction, and the generative model admits a Jacobian. The authors define a series of strenghtenings of identifiability with respect to how accurate to “ground truth” the estimated values are. They then study the effect of sparsity. Finally, they define a diversity condition that allows for element-wise recovery. They evaluate their theoretical results empirically.

**Strengths:**

The main strength of the paper is in abstracting sufficient conditions for recovery of various consistent dictionaries from examples. It is powerful to reason about these properties in a general class of generative models using only set theoretic terms.

**Weaknesses:**

Some of the theoretical claims are imprecise. In particular, I am really puzzled by the interplay between Definition 6 and Theorem 1. Is Theorem 1 a result about methods that satisfy the regularization condition? Or should I read it as a result about the nature of identifiable models, i.e., they are ones that have higher sparsity than all others that are observationally equivalent? This requires clarification. Also, I am a bit confused about Theorem 2: if all the conditions of Theorem 1 are met, you get both generalized and structure identifiability? There are no additional conditions?

Related to all of this: in section 3.3, what is the quantifier over $\hat{z}, \hat{g}$? I thought it was the estimated value. I don’t see where it shows up in Assumption 1 until the last part, and in Theorem 1, the hat versions are specified but somehow you are invoking Defn 6, in which the quantifier is “any” theta?

On the necessity side: Are there any necessity conditions for set-theoretic indeterminacy or generic identifiability?

Finally, it would be helpful in your exposition to map your abstracted properties to properties in general linear dictionary learning, non-linear dictionary learning, and combinatorial dictionary learning. (e.g., diversity = RIP = well-structuredness).

I can consider increasing my evaluation of the paper if the theoretical claims are made more precise.

**Questions:**

In addition to the questions above:

Maybe I’m missing something, but why is Theorem 1, condition (ii) called “regularization”? To me regularization is algorithmic, whereas this is structural.

Can you please clarify why your experiments are about structural properties of dictionary learning, rather than the specific methods you have implemented?

---

> ### Author Response · Authors · 2025-11-22
> **We sincerely appreciate your time and thoughtful feedback (1/3）**
>
> Dear Reviewer xMy9,
>
> We sincerely appreciate your time and thoughtful feedback. We are glad that you found our contributions powerful. In light of the suggestions, we have added **expanded discussions** in the updated manuscript. Please find our detailed point-by-point responses below.
>
> ---
>
> **Q1**: In particular, I am really puzzled by the interplay between Definition 6 and Theorem 1. Is Theorem 1 a result about methods that satisfy the regularization condition? Or should I read it as a result about the nature of identifiable models, i.e., they are ones that have higher sparsity than all others that are observationally equivalent?
>
>
> **A1**: Thank you so much for raising this point. We fully agree with you that it is important to add further highlights to avoid potential confusion. Theorem 1 is a result about the *identifiability of the underlying data generating process*, and does not assume sparsity of the true model. As discussed in **L324-327**, the sparsity term in Condition (ii) applies **only to the estimated model** as a standard regularization during estimation. It does not assume or imply that the ground truth model is sparse.
>
> Definition 6 gives the standard notion of identifiability: it characterizes the allowable ambiguity between observationally equivalent models. The role of the sparsity regularizer is simply to select, among all observationally equivalent estimators, a representative whose dependency structure is the most sparse. This choice lies entirely on the estimation side and does not constrain the form of the true data-generating mechanism.
>
> The interplay is therefore clear. Definition 6 specifies the identifiability class. The sparsity penalty ensures that any observationally equivalent estimator remains within this class and resolves the remaining degrees of freedom in a principled way. This separation between identifiability of the model class and regularization applied to the estimator follows the same pattern used in prior identifiability work, such as nonlinear ICA and related latent variable models [1,2,3]. We have further highlighted these in the updated manuscript (**L328-332**) according to your great suggestion. Please let us know if anything is still not fully clear.
>
> [1] On the identifiability of nonlinear ICA: Sparsity and beyond. NeurIPS 2022
>
> [2] Disentanglement via mechanism sparsity regularization: A new principle for nonlinear ICA. CLeaR 2022
>
> [3] Counterfactual generation with identifiability guarantees. NeurIPS 2023
>
> ---
>
> **Q2**. Also, I am a bit confused about Theorem 2: if all the conditions of Theorem 1 are met, you get both generalized and structural identifiability? There are no additional conditions?
>
> **A2**. Thanks for your question. Yes, that is correct. Under the conditions of Theorem 1, both generalized identifiability and structure identifiability (up to permutation) follow. There are no extra assumptions required. We stated them as separate theorems because they highlight two different aspects of the latent generative process: Theorem 1 emphasizes generalized identifiability of the latent variables, while Theorem 2 focuses on the structure between latent and observed variables. Depending on the downstream task, readers may care about one more than the other, so we present them separately for clarity.
>
>
> ---
>
> **Q3**. Related to all of this: in section 3.3, what is the quantifier over $\hat{z}$, $\hat{g}$? I thought it was the estimated value. I don’t see where it shows up in Assumption 1 until the last part, and in Theorem 1, the hat versions are specified, but somehow you are invoking Defn 6, in which the quantifier is “any” theta?
>
> **A3**. Thank you for the question. Throughout the theorems, when we say “consider two models,” the quantifier is universal: we mean any two models. We have made this explicit in the revision. The remaining conditions then restrict which pairs are compared, for example, requiring them to be observationally equivalent, and incorporating the standard sparsity ($\ell_0$) regularization used during estimation. This regularization applies only to the estimator, not to the data-generating process.

---

> > ### Author Response · Authors · 2025-11-22
> > **We sincerely appreciate your time and thoughtful feedback (2/3）**
> >
> > **Q4**. On the necessity side: Are there any necessity conditions for set-theoretic indeterminacy or generic identifiability?
> >
> > **A4**. Thanks for the insightful question. Certain conditions are necessary, such as injectivity, which ensures that the information is recoverable in principle. This is why similar injectivity assumptions (e.g., RIP-type conditions) appear throughout the prior literature. For our main structural diversity condition, we do not claim necessity. Based on the broader nonparametric identifiability literature, it is likely that if such structural conditions fail, additional sources of information would be required, such as interventions, multiple environments, or restricted function classes; otherwise, classical results already show non-identifiability in general nonlinear latent models [1,2].
> >
> > [1] Nonlinear independent component analysis: Existence and uniqueness results. Neural Networks, 1999
> >
> > [2] Challenging common assumptions in the unsupervised learning of disentangled representations. ICML 2019
> >
> >
> >
> > ---
> >
> >
> >
> > **Q5**. It would be helpful in your exposition to map your abstracted properties to properties in general linear dictionary learning, non-linear dictionary learning, and combinatorial dictionary learning. (e.g., diversity = RIP = well-structuredness).
> >
> > **A5**. Thanks for the great suggestion. We fully agree that a more detailed discussion will definitely be very helpful for readers. In addition to the existing discussion (e.g., **L102-112**, **L378-395**), we have added a dedicated section (**Appendix B.2, L1350-1381**) in the updated manuscript:
> >
> > - **Connection between diversity and sparsity assumptions**. The structural diversity condition is firstly introduced in our work for the nonlinear case, but other conditions on the structure appear in the related field, e.g., the sparsity condition. For instance, the anchor feature assumption [1,2], structural sparsity [3], sparse dictionary [4], and many others. Although both diversity and sparsity concern latent structure, they are fundamentally different. Diversity focuses on variation in the dependency between latent and observed variables, and does not require sparsity at all. It remains valid even in nearly fully connected graphs, as long as there is some variation (e.g., even a single differing edge) in the connectivity patterns across variables. In contrast, sparsity conditions do not imply diversity and always enforce sparse connectivities.
> >
> > - **Connection between injectivity and RIP assumptions**. Injectivity is a standard requirement to ensure that latent information is not lost through the generative map. In the linear setting, this reduces to classical Restricted Isometry Property (RIP) conditions [5,6], which are known to be necessary for linear dictionary learning. Our injectivity condition is the natural nonlinear analogue: it prevents degenerate mappings that collapse latent variation, standard in almost all previous work on nonparametric identifiability [7,8].
> >
> > - **Connection between dependency sparsity and latent sparsity regularizations**. Beyond assumptions on the data-generating process, there are also connections in terms of regularization during estimation. Most prior work on sparse dictionary learning imposes sparsity directly on the recovered latent variables $Z$. The most prominent example is the Sparse Autoencoder (SAE), which is based on sparse dictionary learning and widely used in mechanistic interpretability. However, as highlighted by recent reviews [9], latent sparsity causes issues such as feature absorption and extremely high latent dimensionality (e.g., millions of variables). In contrast, we regularize sparsity on the dependency structure (Jacobian sparsity) rather than the latents themselves, which avoids these issues and has shown benefits both in our own experiments (Sec. 4.2) and in recent work on large language models, such as the Jacobian Sparse Autoencoder [10].
> >
> >
> > [1] Learning topic models–going beyond svd. IEEE Annual Symposium on Foundations of Computer Science, 2012
> >
> > [2] Identifiable variational autoencoders via sparse decoding. TMLR, 2022
> >
> > [3] On the identifiability of nonlinear ICA: Sparsity and beyond. NeurIPS 2022
> >
> > [4] On the uniqueness and stability of dictionaries for sparse representation of noisy signals, IEE Transactions on Signal Processing, 2019
> >
> > [5] Restricted isometry property. A Mathematical Introduction to Compressive Sensing, 2013
> >
> > [6] On the minimax risk of dictionary learning. IEEE Transactions on Information Theory, 2016
> >
> > [7] Identifiability of latent-variable and structural-equation models: from linear to nonlinear. Annals of the Institute of Statistical Mathematics, 2024
> >
> > [8] Towards interpretable deep generative models via causal representation learning. arXiv, 2025
> >
> > [9] Sparse autoencoders find highly interpretable features in language models. arXiv, 2023
> >
> > [10] Jacobian sparse autoencoders: Sparsify computations, not just activations. ICML 2025

---

> > > ### Author Response · Authors · 2025-11-22
> > > **We sincerely appreciate your time and thoughtful feedback (3/3)**
> > >
> > > **Q6**. Why is Theorem 1, condition (ii) called “regularization”? To me regularization is algorithmic, whereas this is structural.
> > >
> > > **A6**. Thank you for the question. We fully agree that regularization is algorithmic. Condition (ii) is the sparsity regularization applied during estimation. It can be enforced through a $L_0$ penalty on the estimated Jacobian, or approximated with an $L_1$ penalty as the common practice [1,2]. This form of regularization has been used in prior work [3,4] and also in recent large-scale models [5]. We follow this established terminology and therefore refer to it as a regularization term. If you feel that an alternative phrasing would help improve clarity, we would be happy to adjust it.
> > >
> > > [1] Sharp thresholds for high dimensional and noisy sparsity recovery using $\ell_1$ constrained quadratic programming (lasso), IEEE Transactions on Information Theory, 2009.
> > >
> > > [2] Local disentanglement in variational auto encoders using jacobian $l_1$ regularization, NeurIPS 2021.
> > >
> > > [3] On the identifiability of nonlinear ICA: Sparsity and beyond, NeurIPS 2022.
> > >
> > > [4] Generalizing nonlinear ica beyond structural sparsity, NeurIPS 2023.
> > >
> > > [5] Jacobian sparse autoencoders: Sparsify computations, not just activations, ICML 2025.
> > >
> > >
> > > ---
> > >
> > > **Q7**. Can you please clarify why your experiments are about structural properties of dictionary learning, rather than the specific methods you have implemented?
> > >
> > > **A7**. Thank you for the question. Our focus is on identifiability theory rather than proposing a specific algorithm. Under the definition of identifiability, the only rigorous way to validate the theory is to examine the correspondence between the ground truth latents and the estimated latents. This is why our experiments target the *properties* proved by the theory, rather than properties of any particular implementation.
> > >
> > > Our evaluations follow the standard metrics used in the identifiability literature:
> > >
> > > - **Figure 4 (generalized identifiability)**: According to the definition, the generalized identifiability concerns set-wise relation between estimated and ground-truth latent variables. For instance, it guarantees that, for two groups of observed variables, the intersection of their estimated latent variables $\hat{Z}\_{I_K \cap I_V}$ is disentangled from the other ground-truth latent variables ${Z}_{I_K \Delta I_V}$, which has been shown by the low $R^2$ score between them. The $R^2$ score is a standard metric to evaluate the set-wise relation between estimated and ground-truth latent variables [1,2,3].
> > >
> > > - **Figure 5 (element identifiability)**: According to the definition, the element identifiability concerns element-wise relation between estimated and ground-truth latent variables. For instance, it guarantees that, there exists a one-on-one function between the estimated and ground-truth latents. Similarly, we use MCC to evaluate it, following most previous work in nonparametric identifiability (e.g., [4], and almost every work in the recent surveys [5, 6]).
> > >
> > > We have highlighted this point in the experiment section of the updated manuscript (**L414-416**). Please feel free to let me know if there are any questions, and we would be more than glad to further elaborate.
> > >
> > > [1] Self-Spervised Learning with data augmentations provably isolates content from style. NeurIPS 2021
> > >
> > > [2] Multi-view causal representation learning with partial observability. ICLR 2024
> > >
> > > [3] Subspace identification for multi-source domain adaptation. NerIPS 2023
> > >
> > > [4] Variational Autoencoders and Nonlinear ICA: A Unifying Framework. AISTATS 2020
> > >
> > > [5] Identifiability of latent-variable and structural-equation models: from linear to nonlinear. Annals of the Institute of Statistical Mathematic, 2023.
> > >
> > > [6] Towards interpretable deep generative models via causal representation learning. arXiv, 2025

---

> > > > ### Author Response · Authors · 2025-12-02
> > > > **Summary of the rebuttal**
> > > >
> > > > Dear AC,
> > > >
> > > > Thanks so much for your time and effort. For convenience, we would like to provide a brief summarization of the rebuttal with Reviewer xMy9.
> > > >
> > > > We appreciate that the reviewer found our contributions helpful. The main questions arose from potential confusion about the theory, and we have given clear and detailed explanations that directly address each point.
> > > >
> > > > As the reviewer mentioned:
> > > >
> > > > >*“I can consider increasing my evaluation of the paper if the theoretical claims are made more precise.”*
> > > >
> > > > We believe our responses, together with added clarifications, fully resolve the concerns. We also highlight that the original theoretical statements were already precise, which is reflected in comments from other reviewers, for example:
> > > >
> > > > >*“The writing is very clean and easy to understand (although the math is dense) with associated well made figures backing up the intuition with toy examples which really helped my understanding of the paper.”* (Reviewer KUfT)
> > > >
> > > > Therefore, we believe all concerns have been addressed, both in the original manuscript and during the rebuttal, and the **rating should be increased as the reviewer indicated**.
> > > >
> > > > Sincerely,
> > > >
> > > > Authors of Submission 564

---

### Author Response · Authors · 2025-12-02
**We sincerely appreciate the time, effort, and thoughtful feedback**

Dear Reviewers, AC, and SAC,

We sincerely appreciate the time, effort, and thoughtful feedback you have devoted to the review. Your comments offered constructive suggestions that will help us further strengthen the manuscript. A summary of the discussion is included at the end of each reviewer’s rebuttal and is also given below:

---

**Reviewer xMy9:** We appreciate that the reviewer found our contributions helpful. The main questions arose from potential confusion about the theory, and we have given clear and detailed explanations that directly address each point.

As the reviewer mentioned:

>*“I can consider increasing my evaluation of the paper if the theoretical claims are made more precise.”*

We believe our responses, together with added clarifications, fully resolve the concerns. We also highlight that the original theoretical statements were already precise, which is reflected in comments from other reviewers, for example:

>*“The writing is very clean and easy to understand (although the math is dense) with associated well made figures backing up the intuition with toy examples which really helped my understanding of the paper.”* (Reviewer KUfT)

Therefore, we believe all concerns have been addressed, both in the original manuscript and during the rebuttal, and the **rating should be increased as the reviewer indicated**.

**Reviewer KUfT:** We are glad the reviewer found the problem interesting, the results sound, and the writing easy to follow. We have provided detailed responses to all questions, and the reviewer confirmed that the clarifications, additional discussion, and further experiments strengthened the paper. As a result, they have **raised their score from 6 to 8**.

**Reviewer U1DX:** We are glad the reviewer found the inductive bias helpful, the theory strong, and the framework novel. Their questions focused on clarifying several specific details in the manuscript, not on the originality or soundness of the work. We have provided detailed responses to each point, supported by new experiments and further discussion. We believe all questions have been fully resolved with clear evidence and comprehensive explanations.

**Reviewer P3Ud:** The reviewer’s main and only question concerned the potential impact of the work. In response, we added a dedicated section outlining implications across several domains. Specifically, we highlighted **two overarching benefits** and **five concrete application areas** that directly benefit from our results, supported by clear methodology and representative references. We believe the **question has been fully addressed** and that the **rating should be adjusted accordingly**.

---

We also summarize our main **contributions** below:

- This paper provides a framework for the identifiability of latent variables in general cases. We address what can still be recovered with guarantees when full identifiability is unattainable, and what forms of bias can be universally adopted. We show that in the nonparametric setting without additional information, the intersections, complements, and symmetric differences of latent variables linked to arbitrary observations, together with the latent to observed dependency structure, remain identifiable up to appropriate indeterminacies. When sufficient structural diversity is present, these results further imply full identifiability of all latent variables under conditions that are strictly more general than existing ones. All identifiability gains follow from a simple inductive bias during estimation that can be readily integrated into most models. We validate the theory and demonstrate the benefits of this bias on both synthetic and real data.

Thank you again for your careful consideration of our work. Please let us know if you have any questions.

Best regards,

Authors of Submission 564

---

### Meta-Review · Area_Chair_3ZFj · 2026-01-06

**Summary:**

This paper introduces Diverse Dictionary Learning, a novel theoretical framework addressing identifiability in general nonlinear latent variable models.

Strengths:

- The set-theoretic notion of identifiability is original and well-motivated, relaxing traditional and often unverifiable assumptions.

- The identifiability results are rigorous and strictly more general than prior sparsity-based results. The sufficient diversity condition provides a unifying view of identifiability assumptions in the literature.

- Validated on both synthetic data (verifying the math) and real-world image datasets.

- Reviewers noted the writing was "very clean" with "well-made figures".

Weaknesses:

- One reviewer (P3Ud) was skeptical about the immediate practical impact. While the authors added an extensive discussion, the value may still feel indirect to readers less familiar with identifiability theory.

Overall, this paper makes a substantive theoretical contribution to representation learning and identifiability. While its impact is more conceptual than immediately performance-driven, this is well within the scope of ICLR. Hence I recommend an acceptance.

**Reviewer Concerns:**

Reviewer xMy9:

- Reviewer raised several questions about the preciseness of statements in Theorem 1 & 2, Definition 6, etc., which were addressed in the rebuttal.

- Reviewer also asked about necessity conditions, regularization, connections to various dictionary learning models, for which the authors provided adequate response.


Reviewer KUFT:

- The reviewer thought positively about the paper after reading the rebuttal, appreciating its implications for interpretability researchers. Though, the reviewer were not entirely confident on the math presented and noted their lack of background on older identifiability and sparsity literature.

Reviewer U1DX

- The "sufficient nonlinearity" condition might only hold asymptotically: Addressed. The authors provided argument that this assumption is typically easy to satisfy in finite sample settings.

- Theory assumes $L_0$ but experiments use $L_1$: Addressed by arguing that $L_1$ is the standard proxy and citing previous sensitivity analysis.

- Penalizing Jacobians is expensive: Addressed by offering two standard strategies to reduce cost.

- The reviewer requested a metric targeting the recovery of the Jacobian's support directly: Addressed. The authors ran new experiments reporting Structural Hamming Distance (SHD).

- Theory might not generalize to non-additive noise or non-invertible processes: Addressed. The authors added Appendix B.6 to discuss these edge cases.

- I am not sure how good of a fit is this theoretical endeavor suitable for ICLR: Addressed with the argument that the venue welcomes work on representation learning.

Reviewer P3Ud:

- It is hard to see the potential impact of this work: Addressed. The authors added a dedicated section detailing several application areas.

**Reviewer Scores:**

Reviewer xMy9: Likely raise to 4 or 5 as the reviewer explicitly stated that they can increase score if the theoretical claims are made more precise. However, it is unclear how reviewer would rate the paper after a better understanding of the theory.

Reviewer KUFT: They raised the score to 8.

Reviewer U1DX: May keep as 6 or raise to 7.

Reviewer P3Ud: May raise to 5 if they were satisfied with the list of applications.

---

### Decision · Program_Chairs · 2026-01-26

Accept (Poster)